# The lipid sensor GPR120 promotes brown fat activation and FGF21 release from adipocytes

Tania Quesada-López[1], Rubén Cereijo[1], Jean-Valery Turatsinze[2], Anna Planavila[1], Montserrat Cairó[1], Aleix Gavaldà-Navarro[1], Marion Peyrou[1], Ricardo Moure[1], Roser Iglesias[1], Marta Giralt[1], Decio L. Eizirik[2] & Francesc Villarroya[1]

The thermogenic activity of brown adipose tissue (BAT) and browning of white adipose tissue are important components of energy expenditure. Here we show that GPR120, a receptor for polyunsaturated fatty acids, promotes brown fat activation. Using RNA-seq to analyse mouse BAT transcriptome, we find that the gene encoding GPR120 is induced by thermogenic activation. We further show that GPR120 activation induces BAT activity and promotes the browning of white fat in mice, whereas GRP120-null mice show impaired cold-induced browning. Omega-3 polyunsaturated fatty acids induce brown and beige adipocyte differentiation and thermogenic activation, and these effects require GPR120. GPR120 activation induces the release of fibroblast growth factor-21 (FGF21) by brown and beige adipocytes, and increases blood FGF21 levels. The effects of GPR120 activation on BAT activation and browning are impaired in FGF21-null mice and cells. Thus, the lipid sensor GPR120 activates brown fat via a mechanism that involves induction of FGF21.

[1] Departament de Bioquímica i Biomedicina Molecular, Institut de Biomedicina, Universitat de Barcelona (IBUB) and CIBER Fisiopatologia de la Obesidad y Nutrición, Avda Diagonal 643, 08028 Barcelona, Spain. [2] ULB Center for Diabetes Research, Medical Faculty, Université Libre de Bruxelles, Avenue Franklin Roosevelt 50, 1050 Brussels, Belgium. Correspondence and requests for materials should be addressed to F.V. (email: fvillarroya@ub.edu).

Brown adipose tissue (BAT) is the main site of non-shivering thermogenesis in mammals. It confers a unique mechanism for energy expenditure and heat production in response to cold and provides a protective mechanism against excessive body weight accumulation in response to overfeeding[1,2]. The interest in brown fat activity as a mechanism of protection against the obesity and metabolic diseases has been renewed by the recent recognition that adult humans possess active BAT, and its activity is negatively associated with obesity and type II diabetes[3]. Many aspects of the molecular mechanisms underlying the function of BAT are known, but we do not comprehensively understand how BAT activity is controlled and integrated with whole organism metabolism to ensure that metabolic substrates are burned and heat is provided. Recent studies unravelled an additional BAT-related means to control energy expenditure, wherein white adipose tissue (WAT) has the capacity to acquire BAT-like properties via the so-called 'browning' process. During this process, sustained thermogenic activation leads to the appearance of the so-called beige or brite adipocytes in WAT depots, which, like classical brown adipocytes, express uncoupling protein-1 (UCP1) and perform uncoupled mitochondrial respiration[4,5]. Several lines of evidence suggest that the browning process is especially relevant in controlling whole-body energy balance[4]. This may reflect its high inducibility in response to environmental factors and the ability of beige cells to use additional, non-UCP1-mediated energy expending mechanisms[6].

Studies aimed at assessing how BAT responds to cold can improve our understanding of the processes that mediate BAT activation. Transcriptomic profiling of BAT from cold-exposed mice can provide a snapshot of how BAT responds to the thermogenic activation and may offer an unbiased look at novel BAT activity-related actors. Recently, RNA sequencing (RNA-seq) has emerged as the best tool for transcriptomic studies, as it does not require a priori knowledge of targets, and shows both high reproducibility and a low frequency of false positives[7–9]. Moreover, RNA-seq can identify 25–75% more genes than complementary DNA (cDNA) microarrays, and it allows assessment of both whole genes and splice variants[10,11].

Here we used RNA-seq to analyse the responsiveness of BAT to the cold-induced thermogenic activation. Departing from these data set, we identified a novel pathway through which thermogenic activation of BAT and the browning of WAT occur via the activation of GPR120 (FFAR4). GPR120 is a G-protein-coupled receptor that binds unsaturated long-chain fatty acids and their derivatives[12]. GPR120 is known to contribute to mediating the anti-inflammatory actions of polyunsaturated fatty acids (PUFAs) and in protecting against obesity and type II diabetes[13,14]. Here we identify a novel pathway of thermogenic regulation through that PUFA-mediated GPR120 activation induces BAT activity and WAT browning via the hormonal factor fibroblast growth factor-21 (FGF21).

## Results

**Effect of cold exposure on BAT transcriptome.** RNA-seq data were obtained from BAT samples obtained from mice housed under thermoneutral conditions or following a 24-h exposure to 4 °C. Among the 21,391 genes detected by the RNA-seq as being 'expressed' (RPKM > 0) a total of 3,470 (16.2%) were significantly modified under the cold condition: 2,498 and 972 were upregulated and downregulated, respectively. To validate our analysis and identify novel candidate genes related to the thermogenic activation in BAT, the top 10% most cold-induced genes were arbitrarily selected for manually curated analysis. Most were already known to be upregulated by cold; these

included key components of mitochondrial uncoupling (UCP1, 4.5-fold induction), lipid metabolism (Elovl3, 9.9-fold induction; and glycerokinase, 5.4-fold induction), intracellular regulation (Dio2, 4.8-fold induction; and PGC-1α, 3.3-fold induction) and extracellular regulation (Bmp8b and FGF21, > 10-fold inductions). Among the top-induced genes that had not been previously studied, we focused on FFAR4 (GPR120). This G-protein-coupled receptor binds unsaturated long-chain fatty acids and their derivatives, and has been proposed to mediate multiple metabolic effects, including anti-inflammation and amelioration of insulin resistance[12,13]. Allelic variants causing loss-of-function put human individuals at risk to develop obesity[14]. A comparison of our RNA-seq data with two microarray-based data sets[7,8] and a digital gene expression profiling[15] revealed that GPR120 was consistently and strongly upregulated in BAT following cold exposure. Here we set out to assess the regulation and function of GPR120 in relation to BAT activation.

**Thermogenic activation upregulates GPR120 in fat depots.** The expression of GPR120 transcript was determined in adipose depots from mice reared at 21 °C in comparison with small intestine and colon that express functional levels of GPR120 (refs 16,17). The highest expression was found in interscapular BAT (iBAT), while specific WAT depots (inguinal, iWAT; epididymal, eWAT; and mesenteric WAT) showed lower but still relevant expression (Fig. 1a; Supplementary Fig. 1). The expression of GPR120 messenger RNA (mRNA) was strongly induced in BAT of mice subjected to either short- or long-term cold exposure (Fig. 1b). Cold also induced GPR120 mRNA expression in iWAT. GPR120 protein levels were increased in both BAT and iWAT after cold exposure (Fig. 1b).

Similar to markers of brown adipocyte thermogenic activity (for example, UCP1), GPR120 was preferentially expressed in mature, differentiated, brown adipocytes rather than in the stromal vascular fraction (Fig. 1c).

In cultured precursor cells from iBAT, GPR120 expression was low at the pre-adipocyte stage, increased progressively during brown adipocyte differentiation and peaked at full differentiation (day 10, maximal expression of UCP1; (Fig. 1d). Norepinephrine or cyclic AMP (cAMP), the major mediators of thermogenic induction, significantly upregulated GPR120 transcript (Fig. 1e) and protein levels (Fig. 1f) in brown adipocytes. The effects of norepinephrine (NE) and cAMP were blunted by the p38 MAPK inhibitor SB202190, but not by the protein kinase-A (PKA) inhibitor H89 (Fig. 1g). This contrasts with UCP1 mRNA whose induction by NE and cAMP was partially blunted both by SB202190 and H89, in accordance with the dual involvement of p38 MAPK and PKA in the control of UCP1 gene expression[18] (Supplementary Fig. 1). GW7647, a peroxisome proliferator-activated receptor alpha (PPARα) activator, did not significantly alter GPR120 expression (Fig. 1g).

In summary, we herein identified GPR120 as a novel component of the acquisition of the differentiated phenotype of brown adipocytes and showed that it is induced in vivo and in vitro in BAT by noradrenergic regulators of thermogenic activation.

**GW9508 increases the thermogenic activity of BAT and WAT.** Mice were treated with GW9508, an activator of GPR120, via their food during 7 days. This treatment did not significantly modify body weight or food intake. Metabolic profiling revealed unaltered glycaemia and triglyceridemia, but GW9508 induced a reduction in insulin levels (Supplementary Table 1), which could reflect improvement in insulin sensitivity. The other tested hormone levels were unaltered following GW9508 treatment

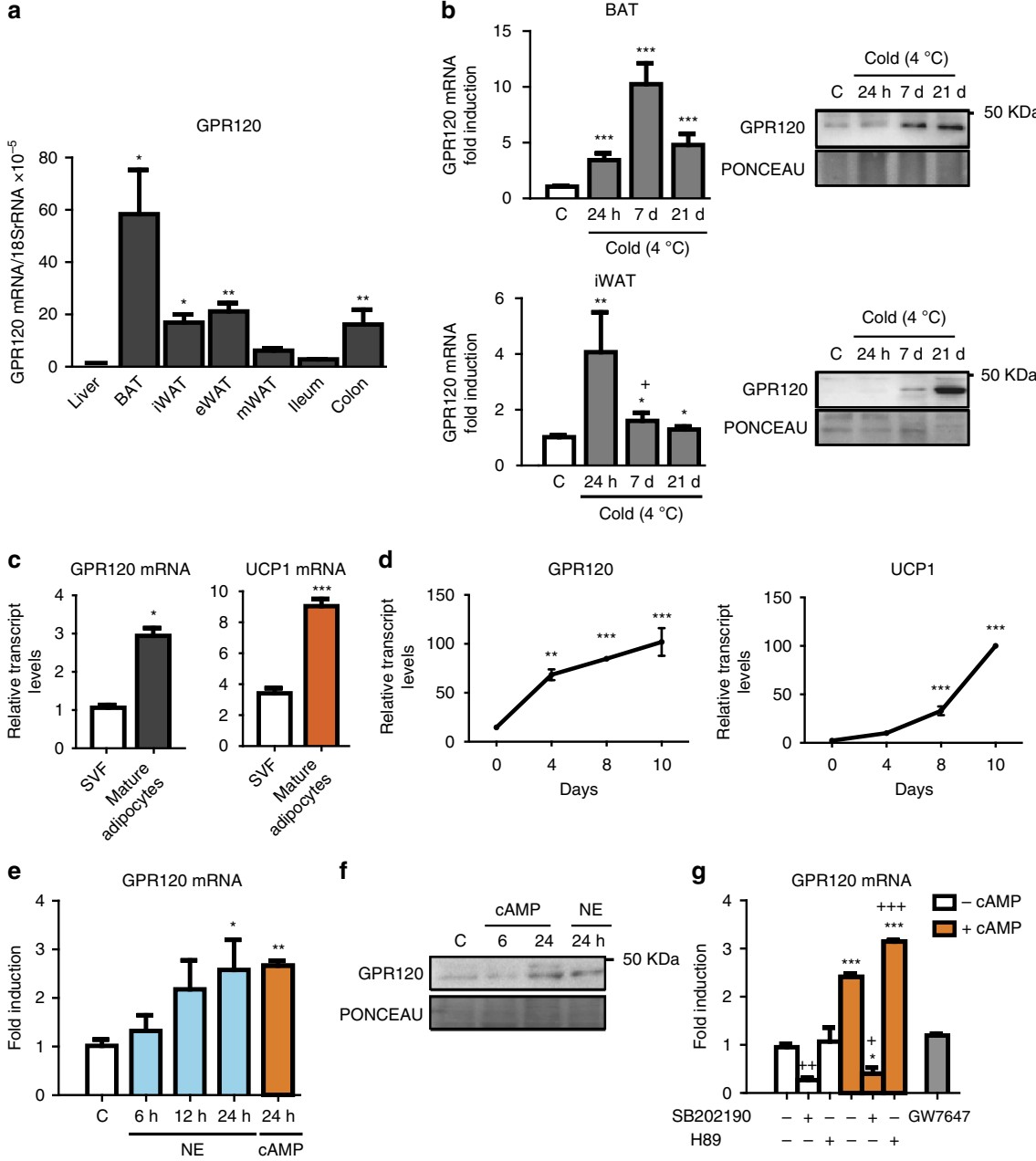

**Figure 1 | Regulation of GPR120 gene expression in BAT and brown adipocytes.** For **a**, **b**, tissues from adult mice were analysed. (**a**) Relative expression of *GPR120* mRNA in liver, BAT, iWAT, eWAT, mWAT, ileum and colon ($n = 4$) was quantified. (**b**) *GPR120* mRNA expression in BAT and iWAT from adult mice maintained at thermoneutrality (29 °C) or exposed to cold (4 °C) for 24 h, 7 days and 21 days ($n = 5$) in the left, representative immunoblot of three independent assays of the relative changes of GPR120 protein, in the right. (**c**) mRNA levels of *GPR120* and *UCP1* in the stromal vascular fraction (SVF) and mature adipocytes obtained from iBAT ($n = 3$). For d-g, BAT precursor cells were differentiated. (**d**) mRNA expression patterns for *GPR120* and *UCP1* during brown adipocyte differentiation in primary cultures, as assessed at days 0 (pre-adipocytes) 4, 8, and 10 ($n = 3$). (**e**) *GPR120* mRNA levels in differentiated brown adipocytes treated with 0.5 µM norepinephrine (NE) for 6, 12 and 24 h (blue bars) or with 1 mM dibutyryl-cAMP for 24 h (orange bars; $n = 4$). (**f**) Representative immunoblot of three independent assays of the relative changes of GPR120 protein levels in response to the indicated NE and cAMP treatments. (**g**) Effects of 10 µM SB202190 (a p38 MAPK inhibitor), or 20 µM H89 (a PKA inhibitor) on the upregulation of *GPR120* mRNA in response to 1 mM dibutyryl-cAMP (orange bars), and effects of 1 µM GW7647 (PPARα agonist, grey bar; $n = 4$). Bars are means + s.e.m. (*$P < 0.05$, **$P < 0.01$ and ***$P < 0.001$ compared with corresponding controls, or ileum; +$P < 0.05$, ++$P < 0.01$, +++$P < 0.001$ for the effects of SB202190 or H89; for **a**, **b**, **e** and **g** analysis of variance with Tukey's *post hoc* test was used; for **c** and **d** two-tailed unpaired Student's *t*-test was performed).

(Supplementary Table 1) with the exception of FGF21 levels, which were markedly induced (see below). The 1-week exposure to GW9508 treatment did not significantly modify the gross masses of iBAT, iWAT, eWAT or mesenteric WAT (Supplementary Table 1). Gene expression analysis revealed significant upregulations among markers of thermogenic activation

in BAT, such as *UCP1*, *PGC-1α*, *CoxIV* and *Sirt3*, as well as *Glut1*, but no change in overall adipogenesis. (Fig. 2a). UCP1 protein levels in the iBAT depot were also significantly increased (Fig. 2a). Microscopy examination of iBAT did not reveal major changes (Fig. 2a). In iWAT, we observed strong evidence of GW9508-induced browning, including upregulation of markers of brown-

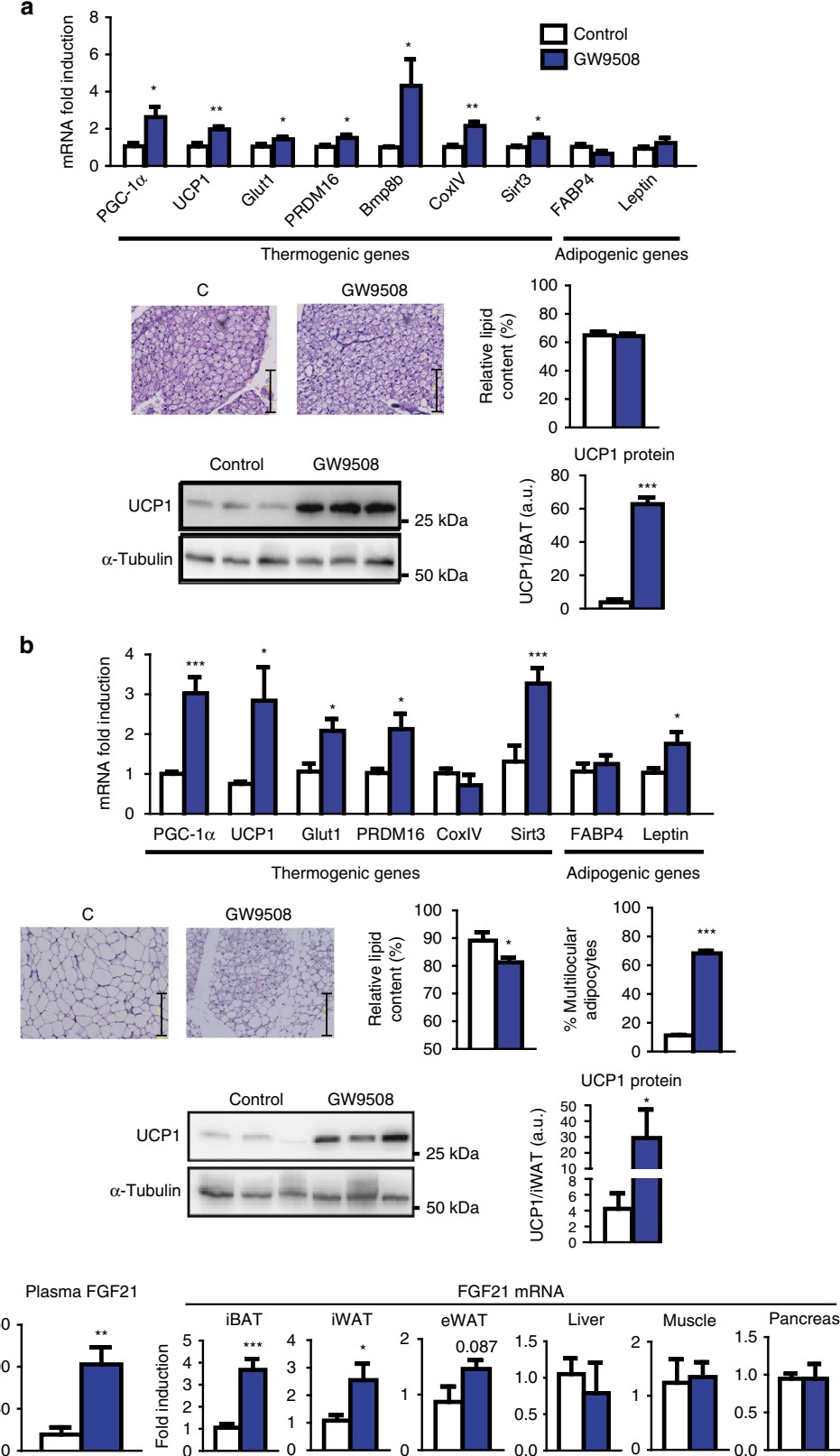

**Figure 2 | GW9508 upregulates thermogenic genes in iBAT and browning in iWAT while inducing FGF21 expression and release.** Adult mice were fed for 7 days a control diet (white bars) or a diet supplemented with GW9508 (blue bars; n = 6). (**a**) Relative expression levels of thermogenic and adipogenic genes in iBAT, representative optical microscopy images from H&E-stained iBAT (scale bar, 125 μm), relative lipid content, UCP1 protein levels and representative UCP1 immunoblot. (**b**) Relative expression levels of thermogenic and adipogenic genes in iWAT, representative optical microscopy from H&E-stained iWAT (scale bar, 125 μm), relative lipid content, percentage of multilocular adipocytes, UCP1 protein levels and representative UCP1 immunoblot. (**c**) Circulating levels of FGF21 protein and *FGF21* mRNA expression levels in iBAT, iWAT, eWAT, liver, skeletal muscle and pancreas. Bars are means + s.e.m. (*P < 0.05, **P < 0.01 and ***P < 0.001 relative to untreated control mice; two-tailed unpaired Student's t-test).

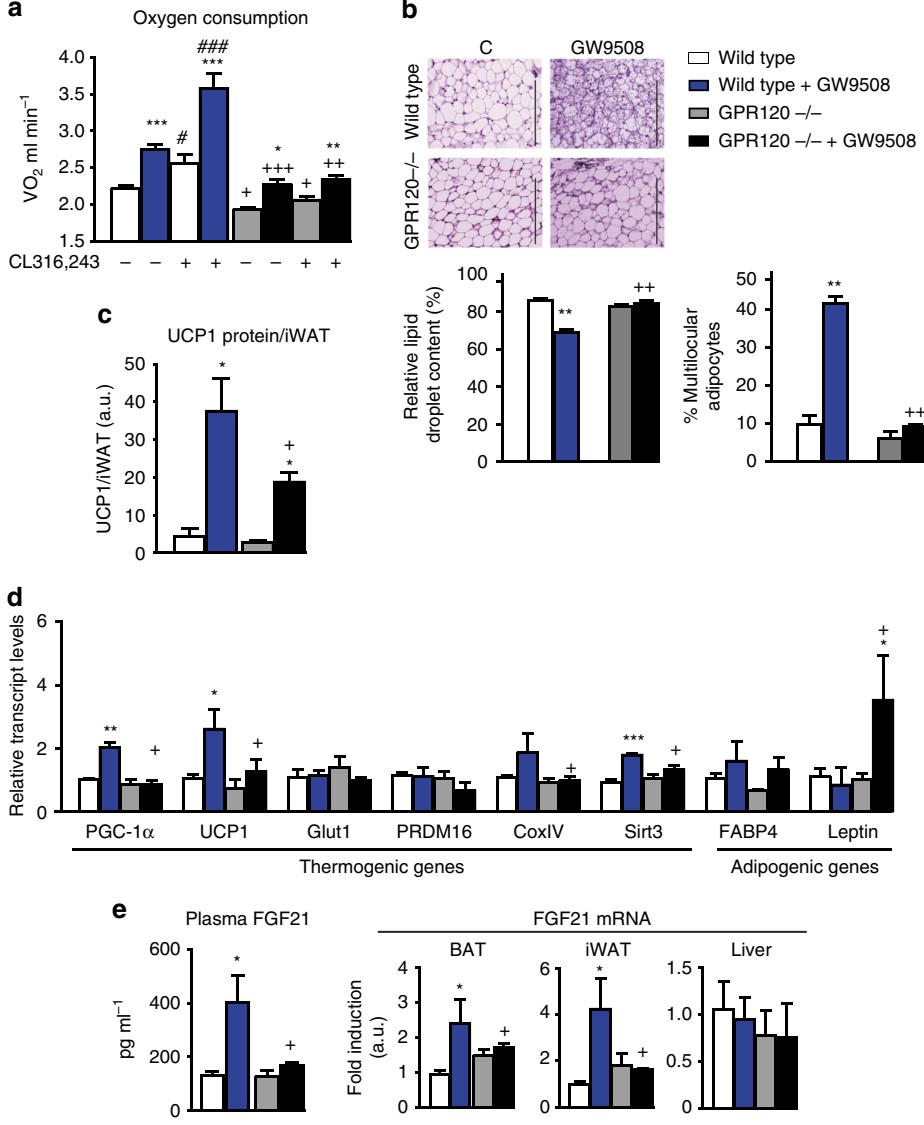

**Figure 3 | GPR120 gene invalidation blunts the effects of GW9508 on adipose tissues and systemic FGF21 levels.** Wild-type and *GPR120 − / −* mice were fed a control diet (white and grey bars, respectively) or a diet supplemented with GW9508 (blue and black bars, respectively) for 7 days (*n* = 5 per group). (**a**) Oxygen consumption in basal conditions and after CL316,243 injection. (**b**) Representative optical microscopy from H&E-stained iWAT (scale bar, 200 μm), relative lipid content and percentage of multilocular adipocytes. (**c**) UCP1 protein levels in iWAT. (**d**) Relative expression levels of thermogenic and adipogenic genes in iWAT. (**e**) Circulating levels of FGF21 protein and *FGF21* mRNA expression levels in iBAT, iWAT and liver. Bars are means + s.e.m. (*$P < 0.05$, **$P < 0.01$ and ***$P < 0.001$ relative to untreated control mice of each genotype; +$P < 0.05$, + +$P < 0.01$ and + + +$P < 0.001$ for the comparisons between genotypes under same GW9508 treatment status; #$P < 0.05$, ###$P < 0.001$ for the effects of GW9508 treatment; analysis of variance with Tukey's *post hoc* test).

related thermogenesis (*UCP1*, *PGC-1α* and *Sirt3*), and the beige adipogenesis-related gene *PRDM16* (Fig. 2b). Consistent with these data, GW9508-treated mice exhibited multiple multilocular adipocytes in iWAT, which are typically associated with the browning process (Fig. 2b). UCP1 protein levels in iWAT were significantly induced by GW9508 (Fig. 2b). Epididymal WAT showed some signs of browning, such as upregulation of *UCP1*, *PGC-1α* and *Sirt3*, but no multilocular beige adipocytes (Supplementary Fig. 2).

As FGF21 is induced in association with the thermogenic activation of BAT and browning of WAT[19,20], we examined FGF21 in treated mice. We found that circulating levels of FGF21 were strongly increased in GW9508-treated mice, as were the *FGF21* mRNA expression levels in BAT and iWAT (Fig. 2c), but not in other tissues such as liver, skeletal muscle or pancreas. The

lack of effects of GW9508 treatment on hepatic *FGF21* gene expression suggested that the upregulation of circulatory FGF21, following GW9508 treatment was not due to a hepatic effect. In addition to activating GPR120, GW9508 can activate another G-protein-coupled receptor, GPR40, which is expressed in the intestine[21] but absent in adipose tissues and minimally expressed in hepatic cells[17,22]. Accordingly, GPR40 transcript expression was undetectable in BAT and most WAT depots (Supplementary Fig. 3a). Moreover, the intestine does not express FGF21 under basal conditions[23] or following GW9508 treatment (Supplementary Fig. 3b). The glucagon gene (encoding glucagon-like peptide (GLP)-1), a target of GPR120 in the intestine[21], was not altered by GW9508 treatment (Supplementary Fig 3b). These results indicate that GW9508 treatment has only minor intestinal effects under this experimental setting.

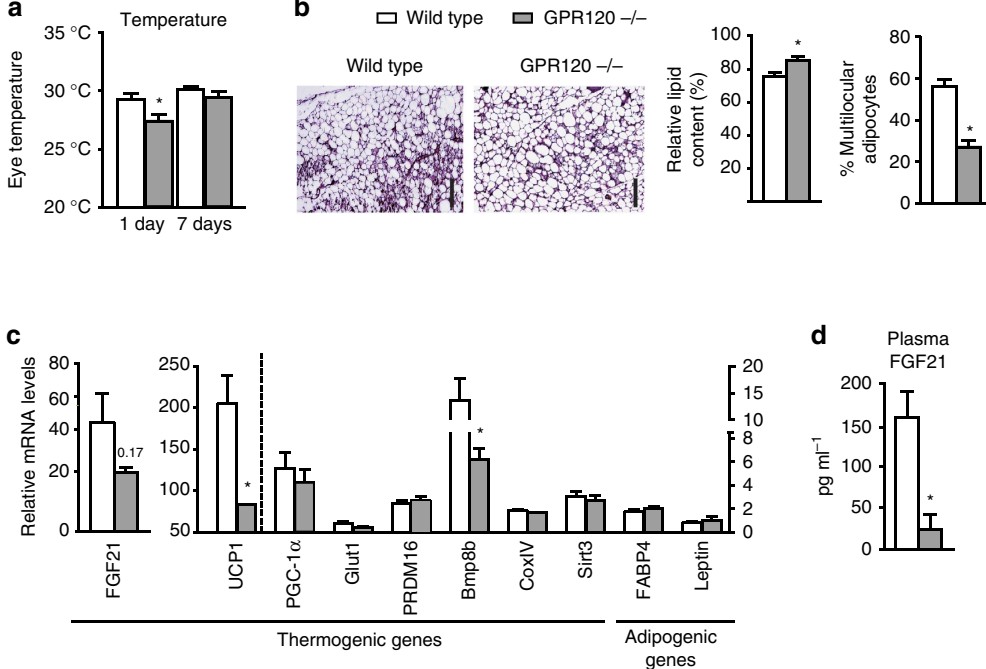

**Figure 4 | GPR120 gene invalidation compromises thermoregulation and iWAT browning in association with a reduction in FGF21 levels.**
Wild-type (Wt, white bars) mice and *GPR120 − / −* mice (grey bars) mice were exposed to cold (4 °C) for 7 days (*n* = 5). (**a**) Body temperature on days 1 and 7 of cold exposure. (**b**) Representative images of H&E-stained iWAT (scale bar, 200 µm), the relative lipid droplet content, and the percentage of multilocular adipocytes. (**c**) Relative expression levels of thermogenic and adipogenic genes in iWAT. (**d**) Circulating levels of FGF21. Bars are means + s.e.m. (*P < 0.05, **P < 0.01 and ***P < 0.001 relative to wild-type animals exposed to cold; two-tailed unpaired Student's *t*-test).

**Impaired action of GW9508 in *GPR120-null* mice**. To directly assess the role of GPR120 for the effects of GW9508, we analysed *GPR120*-null mice. Under basal conditions, *GPR120*-null mice did not show any significant change in body weight or in the main metabolic parameters evaluated (for example, glycaemia or triglyceridemia; Supplementary Table 2). This agrees with previous reports[13,14]. To examine the contribution of GPR120 to the actions of GW9508, we treated wild-type and *GPR120*-null mice with the drug for 7 days. The treatment with the drug did not alter glycaemia, triglyceridemia and insulinaemia in *GPR120*-null mice (Supplementary Table. 2). GW9508-treated wild-type mice showed increased oxygen consumption both under basal conditions and following the injection with the β3-adrenergic agonist CL316,243 (Fig. 3a), consistent with the enhanced iWAT browning and signs of BAT activation shown previously. In contrast, *GPR120*-null mice showed a reduction in basal and CL316,243-triggered oxygen consumption. GW9508 treatment increased oxygen consumption in *GPR120*-null mice, but the attained levels were lower in *GPR120*-null versus wild-type mice. Moreover, the capacity of CL316,243 to induce oxygen consumption was strongly impaired in GW9508-treated *GPR120*-null mice (Fig. 3a). *GPR120*-null mice treated with GW9508 exhibited iBAT with larger lipid droplets, which is reminiscent of decreased thermogenic activity (Supplementary Fig. 4). Expression levels of thermogenesis-related transcripts was significantly altered in *GPR120*-null mice for *PGC-1α* and, especially, for *Bmp8b*, with decreased GW9508-triggered upregulations in *GPR120*-null versus wild-type mice. Total levels of UCP1 protein were unaltered in *GPR120*-null mice (Supplementary Fig. 4).

Browning, identified by the presence of numerous multilocular adipocytes and the induction of several brown-versus-white-related genes (for example, *PGC-1α*, *UCP1* and *Sirt3*) was

enhanced in the iWAT of GW9508-treated wild-type mice, but this induction was markedly impaired in *GPR120*-null mice (Fig. 3b,d). UCP1 protein levels in iWAT were strongly increased in wild-type mice in response to GW9508 treatment, whereas GW9508-treated *GPR120*-null showed significantly lower levels of UCP1 protein (Fig. 3c).

GPR120 invalidation had especially marked effects on circulating FGF21 levels and *FGF21* expression in the iBAT and iWAT of GW9508-treated mice. The increase of plasma FGF21 observed in GW9508-treated wild-type mice was abrogated in *GPR120*-null mice, as was the significant induction of FGF21 transcript levels elicited by GW9508 in BAT and iWAT (Fig. 3e). There was no alteration of *FGF21* gene expression in the liver, regardless of the treatment or mouse genotype.

These data indicate that GPR120 is largely required for the effects of GW9508 in BAT and iWAT '*in vivo*', although the additional involvement of GPR40 cannot be ruled out in light of the dual agonist properties of GW9508.

**Impaired thermogenesis and browning in *GPR120*-null mice**. We next tested the effects of cold exposure on *GPR120*-null mice. Most *GPR120*-null mice tolerated exposure to cold (4 °C), but ∼20% developed hypothermia (a 10 °C or more drop in body temperature) within the first 24 h of cold exposure; none of the wild-type mice developed similar hypothermia. The estimated core temperature was significantly lower in *GPR120*-null mice than in wild-type mice after 1 day of cold exposure and tended to remain lower after 7 days of cold exposure (Fig. 4a). The microscopic morphology of iBAT did not show major alterations in cold-exposed *GPR120*-null mice and thermogenic gene expression pattern was essentially unaltered in *GPR120*-null mice relative to wild-type controls after 7 days of cold (Supplementary Fig. 5). However, *leptin* expression was increased in *GPR120*-null mice, which is consistent with some extent of 'whitening' in

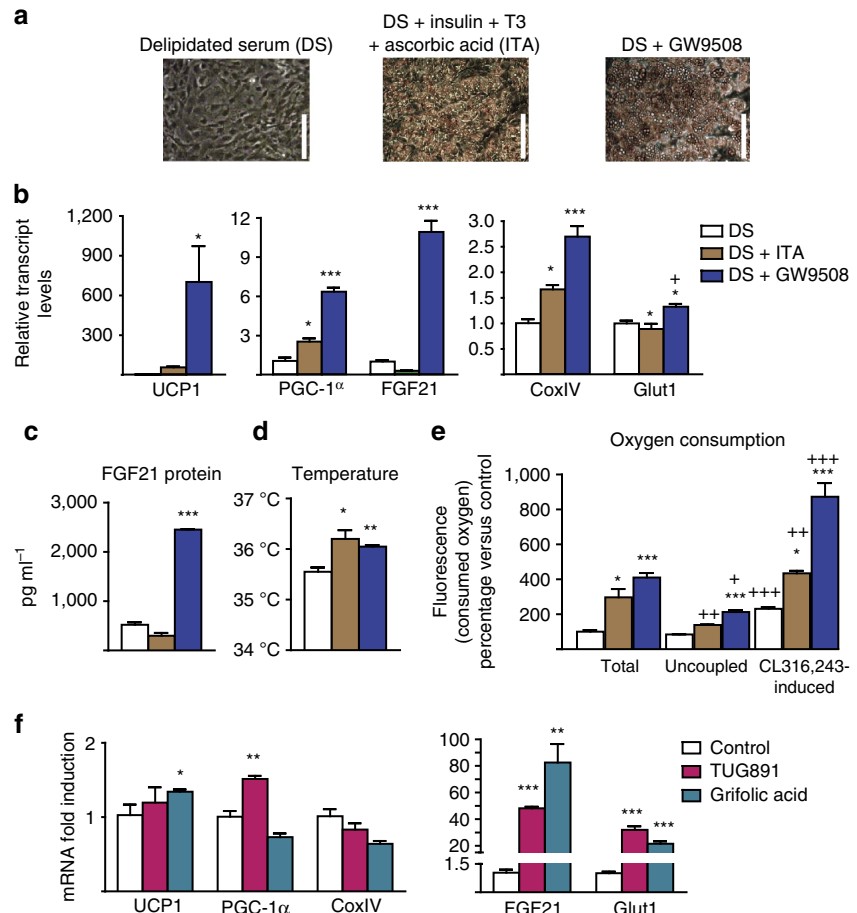

**Figure 5 | GPR120 activation induces brown adipocyte differentiation and increases FGF21 expression and release.** For **a–e**, iBAT precursors were differentiated in the presence of standard culture medium supplemented as follows: with 10% delipidated serum (DS, white bars; $n = 4$); with DS plus insulin, triiodothyronine and ascorbic acid (DS + ITA, brown bars; $n = 5$); or with DS plus 100 µM GW9508 (blue bars; $n = 4$). (**a**) Representative optical microscopy images from cells at the end of the differentiation (day 9) (scale bars, 200 µm). (**b**) Relative mRNA expression levels of *UCP1*, *PGC-1α*, *COXIV*, *FGF21* and *Glut1*. (**c**) FGF21 protein levels in cell culture media (4 day accumulation). (**d**) Cell culture temperature. (**e**) Total and uncoupled (oligomycin-resistant) respiration, and respiration after CL316,243 treatment (**f**) iBAT precursors from mice were differentiated with DS + ITA, mRNA expression levels of *UCP1*, *PGC-1α*, *CoxIV*, *FGF21* and *Glut1* after 24 h treatment with TUG-891 (200 µM, pink bars) or grifolic acid (100 µM, turquoise bars). Bars are means + s.e.m. (*P* values: \**P* < 0.05, \*\**P* < 0.01 and \*\*\**P* < 0.001 versus DS (**b–e**) or versus controls (**f**); + *P* < 0.05, + + *P* < 0.01 and + + + *P* < 0.001 for uncoupled respiration or induction in respiration upon CL316,243 versus total respiration; analysis of variance with Tukey's *post hoc* test).

BAT (Supplementary Fig. 5). iWAT browning was significantly impaired in cold-exposed *GPR120*-null mice. There were far fewer clusters of multilocular adipocytes in the iWAT of *GPR120*-null mice compared with wild-type controls (Fig. 4b), whereas the relative lipid content was higher. The expression levels of the thermogenic genes *UCP1* and *Bmp8b* were reduced, and *FGF21* gene expression tended to be lower (Fig. 4c). Circulating FGF21 levels were significantly reduced in cold-exposed *GPR120*-null mice (Fig. 4d), whereas FGF21 transcript expression in the liver and skeletal muscle was unaltered (Supplementary Fig. 5).

**GPR120 activation induces brown adipocyte thermogenesis**. We next used cell culture systems of brown and beige adipocytes to assess whether the abilities of GPR120 activation to promote BAT activation, WAT browning and FGF21 induction were cell autonomous. GW9508 was used to determine the specific effects of GPR120 activation as brown adipocytes (similar to white adipocytes) lack detectable levels of *GPR40* transcript and thus the actions of GW9508 are only attributable to GPR120 activation[13,24].

First, we determined whether GPR120 activation targets brown adipocyte differentiation. Precursor cells were obtained from iBAT stromal vascular fractions and cultured as previously reported[25]. On day 4 of culture, regular medium was replaced to contain 10% delipidated serum, which does not allow differentiation (Fig. 5a)[26]. The addition of insulin, triiodothyronine and ascorbic acid (ITA) to the culture was sufficient to induce differentiation. Notably, the addition of GW9508 instead of ITA also yielded a robust differentiation of brown adipocytes (Fig. 5a). Analysis of brown differentiation marker genes (for example, *UCP1*, *PGC-1α*, *COXIV* and *Glut1*) indicated that GW9508 triggered a stronger induction compared with ITA (Fig. 5b). Among the tested genes, the highest GW9508-induced upregulation was seen for FGF21 (Fig. 5b). These effects on *FGF21* gene expression were associated with a strong release of FGF21 to the media of GW9508-treated cultures (Fig. 5c). Highly sensitive thermography[27,28] showed that GW9508 treatment increased the cell cultures temperature (Fig. 5d). Moreover, oxygen consumption in brown adipocytes differentiated in the presence of GW9508 was as high as in ITA-differentiated cells (Fig. 5e). GW9508-induced differentiation also enhanced the oxygen consumption triggered by the β3-adrenergic agonist CL316,243 (Fig. 5e).

We next analysed the effects of GPR120 activation on mature, differentiated, brown adipocytes (day 9 of culture). Treatment

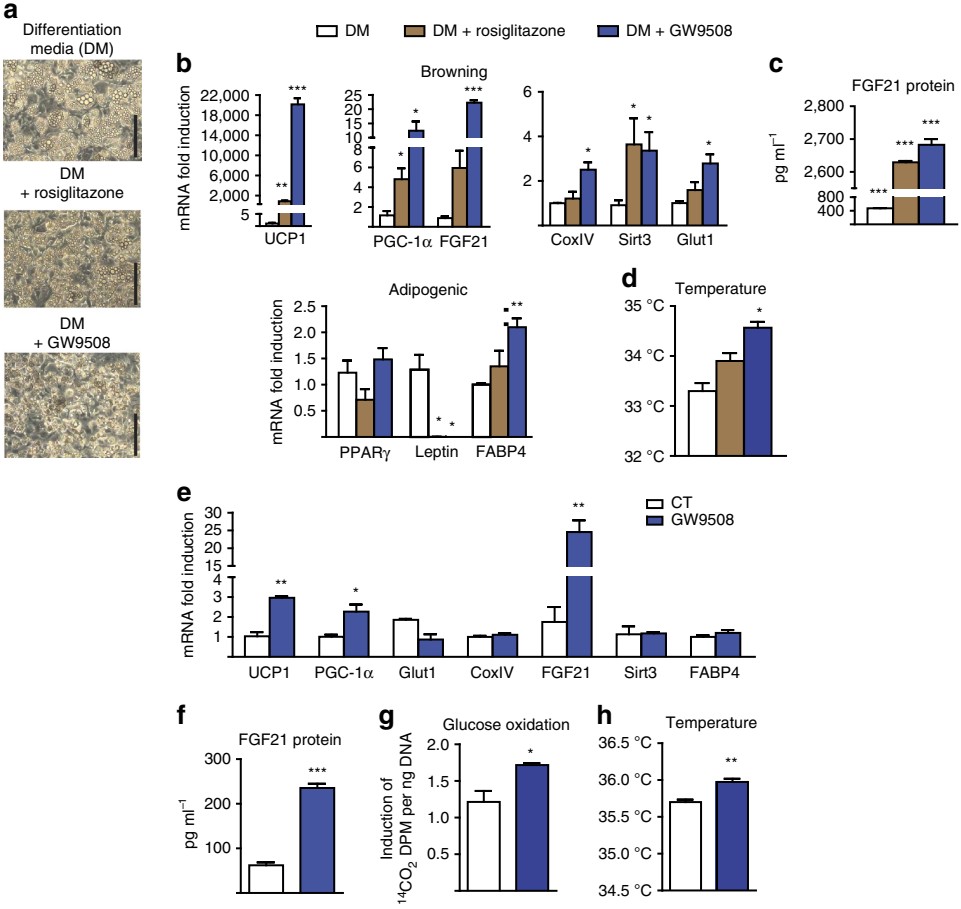

**Figure 6 | GPR120 activation promotes beige adipocyte differentiation and increases FGF21 expression and release.** For **a**–**d**, iWAT precursors from mice were differentiated in the presence of the differentiation media (DM, white bars), supplemented with rosiglitazone to drive beige differentiation (DM + rosiglitazone, brown bars; $n = 4$) or treated with GW9508 instead of rosiglitazone (DM + GW9508, blue bars; $n = 5$; see the Methods section). (**a**) Representative optical microscopy images at the end of differentiation (day 7; scale bar, 200 μm). (**b**) Relative mRNA expression levels of browning-related and general adipogenic genes. (**c**) FGF21 protein levels in the cell culture medium (4 day accumulation). (**d**) Cell culture temperature. For **e**–**h**, iWAT precursors were differentiated and treated during 24 h with GW9508 (100 μM, blue bars; $n = 5$) or nor treated ($n = 3$). (**e**) mRNA expression levels of *UCP1*, *PGC-1α*, *Glut1*, *COXIV*, *FGF21*, *Sirt3* and *FABP4*. (**f**) FGF21 protein levels in culture media (24 h accumulation). (**g**) Glucose oxidation rate. (**h**) Cell culture temperature. Bars are means + s.e.m. (*$P < 0.05$, **$P < 0.01$ and ***$P < 0.001$ versus DM (**b**–**d**) or versus controls (**e**–**h**); for **b**–**d**, analysis of variance with Tukey's *post hoc* test was performed; for **e**–**h**, two-tailed unpaired Student's *t*-test).

with GW9508 for 24 h led to a marked induction of *UCP1*, *Glut1* and *FGF21* expression, as well as a strong increase in the secretion of FGF21 protein (Supplementary Fig 6). Moreover, GW9508 significantly induced the glucose oxidation and tended to increase the cell culture temperature (Supplementary Fig. 6).

To examine whether drugs that also activate GPR120 specifically cause similar effects, we tested TUG-891[29] and grifolic acid[30]. Indeed 0.2 mM TUG-891 and 0.1 mM grifolic acid strongly upregulated *FGF21* and *Glut1*, and had milder effects on *UCP1* and *PGC-1α* (Fig. 5f).

Collectively, these results indicate that GPR120 activation has cell-autonomous effects on brown adipocytes, including a strong effect in promoting brown adipocyte differentiation and a particularly strong activation of FGF21 expression and release associated with signs of the enhanced oxygen consumption, glucose oxidation and thermogenesis.

**GPR120 activation induces beige adipocyte thermogenesis.**
To analyse the cell-autonomous effects of GPR120 activation on the WAT browning process, primary cultures of precursor cells

obtained from iWAT were cultured in regular differentiation medium and exposed to rosiglitazone to promote acquisition of the beige phenotype[31]. Rosiglitazone treatment did not alter the morphology of differentiating cells or the lipid droplet accumulation, but did tend to decrease the lipid droplet size (Fig. 6a). Rosiglitazone did not alter the expression of general adipogenic marker genes in differentiating adipocytes; instead it downregulated leptin (a marker of the WAT versus BAT phenotype) and strongly upregulated the beige markers *UCP1* and *PGC-1α* (Fig. 6b). Supplementation with GW9508 instead of rosiglitazone caused a similar, even stronger, induction of browning, as evidenced by upregulation of the beige marker genes (*UCP1*, *PGC-1α* and *Sirt3*; Fig. 6b) and downregulation of leptin, while leaving most general adipogenic genes unaltered. Rosiglitazone-induced and especially GW9508-induced beige differentiation strongly upregulated the expression (Fig. 6b) and release of FGF21 (Fig. 6c). Consistent with the gene expression data, the heat production was significantly higher in GW9508-treated beige cells (Fig. 6d).

We next questioned whether the effects of the GPR120 activation also occurred in adipocytes that had differentiated to a beige phenotype. Indeed, GW9508 treatment of differentiated

beige adipocytes for 24 h triggered a moderate but significant upregulation of thermogenic marker genes (UCP1 and PGC-1α), and a stronger induction of FGF21 expression (Fig. 6e) and release (Fig. 6f). GW9508 also increased the glucose oxidation rate (Fig. 6g) and cellular heat production (Fig. 6h) in beige adipocytes.

Compared with iWAT precursors, eWAT precursors yielded similar results with GW9508 inducing even stronger browning than rosiglitazone (as shown by gene expression analysis) and powerfully inducing FGF21 expression and release in differentiated adipocytes (Supplementary Fig. 7).

**n-3 PUFAs promote brown fat activation through GPR120.** GPR120 is assumed to interact with unsaturated fatty acids—or derivatives—at the cell surface and trigger subsequent intracellular effects[12]. Since omega-3 PUFAs are more potent than omega-6 or omega-9 PUFAs in eliciting GRP120-mediated effects[12,13], we analysed the effects of α-linolenic (ALA), eicosapentaenoic (EPA) and docosahexaenoic (DHA) acids.

Supplementation of delipidated cell culture medium with EPA or ALA markedly promoted the morphological adipogenic differentiation of brown pre-adipocytes obtained from iBAT, to a degree similar to that elicited by ITA (Fig. 7a). In contrast, DHA had barely significant effects. The thermogenic genes, especially UCP1 and FGF21, were markedly induced by EPA and ALA, relative to ITA. Some induction of general adipogenic genes was also observed, especially in response to EPA (Fig. 7b). EPA and ALA also enhanced FGF21 protein secretion relative to ITA (Fig. 7c). The cellular temperature was increased by EPA relative to cells cultured in delipidated medium, but there was no significant temperature difference relative to ITA-differentiated cells (Fig. 7d). EPA treatment increased oxygen consumption to an extent similar to that elicited by ITA, under both basal and CL316,243-treated conditions (Fig. 7e).

EPA and ALA treatment of beige adipocytes undergoing differentiation from iWAT precursors promoted their morphological differentiation (Fig. 7f), significantly upregulated the tested thermogenic-related genes (with the exception of a non-significant effect of ALA on UCP1 and PGC-1α), and downregulated leptin (Fig. 7g). EPA and ALA increased FGF21 gene expression and release (Fig. 7h), but only EPA significantly increased cell temperature. Overall, in both, the brown and beige experimental settings, EPA tended to be more powerful than ALA.

As PPARα is known to mediate transcriptional effects of fatty acids in brown adipocytes[25] and regulates FGF21 expression in the liver[32,33], we determined the effects of EPA on primary cultures of precursor cells from the iBAT of PPARα-null mice. As expected, the absence of PPARα did not alter the responsiveness of brown adipocytes to GW9508 (Supplementary Fig 8a). EPA still promoted brown adipocyte differentiation and strongly induced FGF21 in cells devoid of PPARα, although the levels of FGF21 expression attained were smaller (Supplementary Fig 8a). We also determined whether the effects of EPA could involve PPARγ. Exposure of differentiating brown adipocytes to the PPARγ inhibitor, GW9662, did not significantly alter the effects of EPA or GW9508 on the morphological differentiation; the significant upregulation of the FGF21 gene in response to EPA was maintained, but the levels achieved were significantly less than those observed in the absence of the PPARγ inhibitor (Supplementary Fig 8b). These results indicate that although PPARs are not necessary for the ability of EPA to induce FGF21, they may contribute to some of the effects of EPA.

We further determined the involvement of GPR120 in mediating the effects of EPA on the brown and beige differentiation of pre-adipocytes. For this purpose, we analysed precursor cells cultures from the iBAT and iWAT of GPR120-null mice and wild-type controls. Precursor cells from GPR120-null iBAT showed much less differentiation into brown adipocytes, as evidenced by decreased acquisition of brown adipocyte morphology (Fig. 8a), impaired expression of thermogenic marker genes (UCP1 and FGF21; Fig. 8b), and signs of reduced expression of general adipogenesis marker genes. EPA treatment significantly induced FGF21 mRNA expression (Fig. 8c) and FGF21 protein secretion (Fig. 8d) in wild-type brown adipocytes, but not in GPR120-null cells that were totally insensitive to EPA. Precursor cells from GPR120-null iWAT showed a delayed differentiation (Fig. 8e; see microscopic morphology on day 3 of culture), but ultimately reached similar levels of thermogenic and adipogenic gene expression by day 7. The abilities of EPA to induce FGF21 mRNA expression and FGF21 protein release were significantly reduced in GPR120-null beige adipocytes (Fig. 8f).

Next, we used two independent approaches to determine whether EPA affects FGF21 expression and release by previously differentiated brown or beige adipocytes, and whether GPR120 mediated these effects. First, we treated differentiated cells with the GPR120 antagonist, AH7614 (ref. 34). As expected, AH7614 significantly inhibited the capacity of GW9508 to induce FGF21 expression and release in brown and beige adipocytes (Fig. 8g). AH7614 also inhibited the effects of EPA on FGF21 expression and release, indicating that these effects involve GPR120 in brown and beige adipocytes.

Second, we performed short interfering RNA (siRNA)-driven knockdown of GPR120 in brown adipocytes subjected to in vitro differentiation. In these cells, we obtained an 80% reduction in the mRNA expression of the GPR120 as compared with controls (Fig. 8h, left). Under these conditions of GPR120 knocking down, the abilities of GW9508 and EPA to induce FGF21 gene expression were also strongly impaired (Fig. 8h). GPR120 knocking down also impaired the induction of UCP1 and PGC-1α by EPA.

Collectively, these findings indicate that EPA induces brown and beige differentiation, and enhances FGF21 gene expression and FGF21 protein secretion mostly via GPR120 activation.

**GPR120-mediated induction of FGF21 involves p38 MAPkinase.** To further explore the mechanisms through which GPR120 activation induces FGF21 gene expression, differentiated brown adipocytes were treated with GW9508 in the presence or not of intracellular kinase inhibitors. U-0128 and wortmannin significantly reduced the upregulation of FGF21 gene expression in response to GW9508 or EPA (Supplementary Fig 9a), confirming the previously reported involvement of ERK1/2 and PI3kinase, respectively, in the intracellular actions following GPR120 activation in other cell systems, such as white adipocytes[13]. In contrast, the inhibitors of PKA or AMP kinase had no effect (Supplementary Fig 9a). Interestingly, the p38 MAPK inhibitor, SB202190, strongly impaired GW9508- or EPA-induced FGF21 gene expression. Treatment of brown adipocytes with GW9508 or EPA induced the phosphorylations of p38 MAPK and (as expected) ERK1/2, but not of other regulatory proteins such as CREB (Supplementary Fig 9b). As p38 MAPK mediates the regulation of FGF21 in response to noradrenergic stimuli in BAT[19], we determined the effects of GW9598 and EPA on the transcriptional regulation of the FGF21 gene promoter. We found that these agents significantly induced the activity of the FGF21 promoter (Supplementary Fig 9c). Conversely, a version of the promoter devoid of the p38 MAPK-responsive site failed to respond to GW9508 or EPA. Moreover, transfection of an expression vector for MKK6 (MKK6-K82A), which acts as dominant negative for p38 MAPK-dependent activation[35], blunted the effects of GW9508 and EPA on the FGF21 gene promoter activity. These findings suggest that GPR120-induced

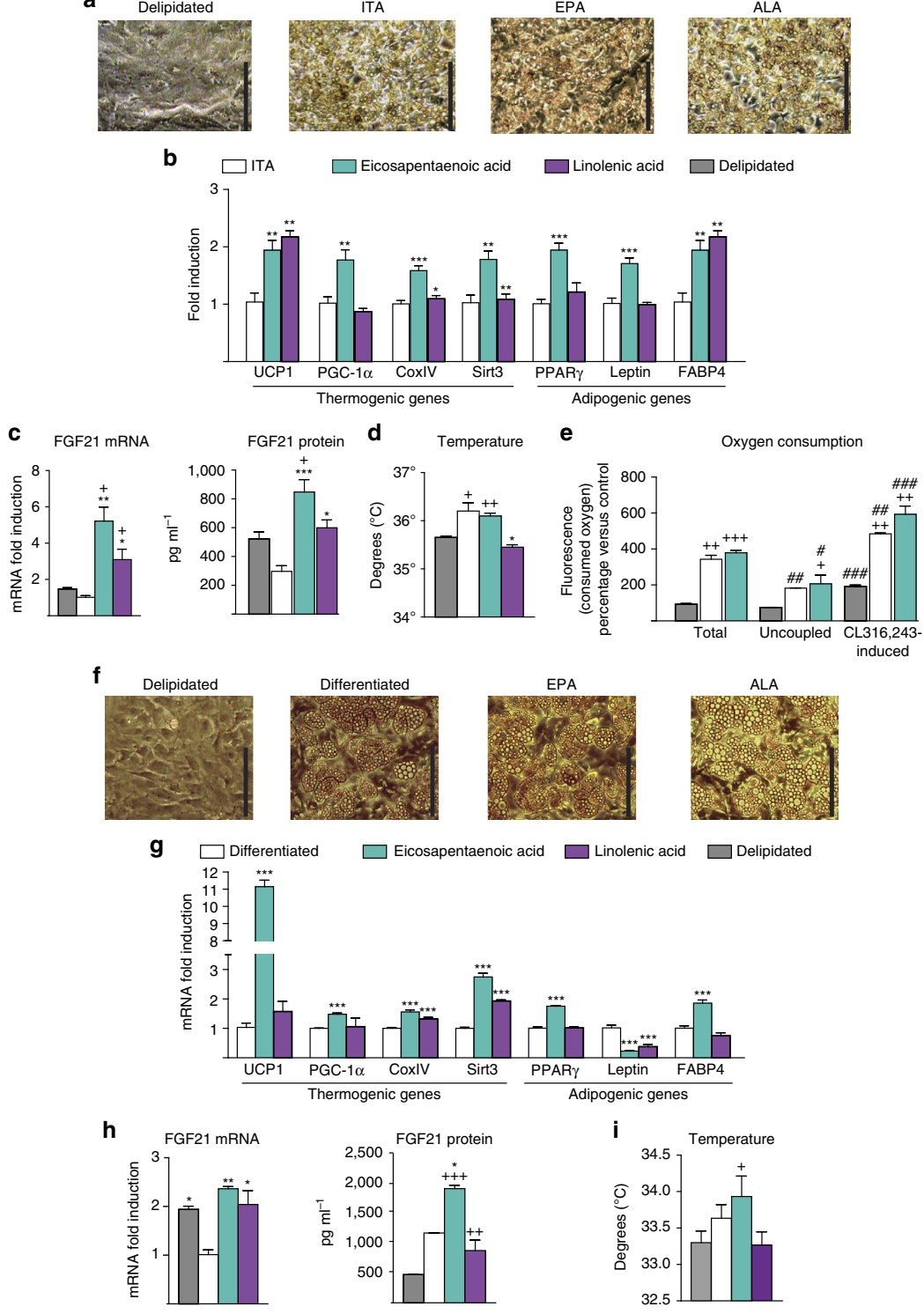

**Figure 7 | n-3 PUFAs upregulate thermogenic genes expression and FGF21 expression and release.** For **a–d**, iBAT precursors were cultured in media containing only delipidated serum (grey bars), or in the presence of insulin, T3 and ascorbic acid (ITA, white bars) or supplemented with eicosapentaenoic acid (EPA, tourquoise bars) or linolenic acid (ALA, purple bars). (**a**) Representative optical microscopy images of cultured brown precursors after the treatment (scale bar, 200 μm). (**b**) Relative mRNA expression of thermogenesis-related and general adipogenesis-related genes. (**c**) *FGF21* mRNA in adipocytes and protein levels in culture media. (**d**) Cell culture temperature. (**e**) Total and uncoupled respiration after differentiation and after treatment with CL316,243. For **f–i**, iWAT precursors were cultured in media containing only delipidated serum (grey bars), adipogenic cocktail (see the Methods section, white bars), or supplemented with EPA (turquoise bars) or ALA (purple bars). (**f**) Representative optical microscopy images of cultured beige precursors after the treatment (scale bar, 200 μm). (**g**) Relative mRNA expression of thermogenesis-related and general adipogenesis-related genes. (**h**) *FGF21* mRNA in adipocytes, protein levels in culture media and (**i**) cell temperature at the end of the treatment. Bars are means + s.e.m.(*$P < 0.05$, **$P < 0.01$ and ***$P < 0.001$ versus differentiated ITA, +$P < 0.05$, ++$P < 0.01$ and +++$P < 0.001$ versus delipidated, #$P < 0.05$, ##$P < 0.01$ and ###$P < 0.001$ relative to total respiration; analysis of variance with Tukey's *post hoc* test).

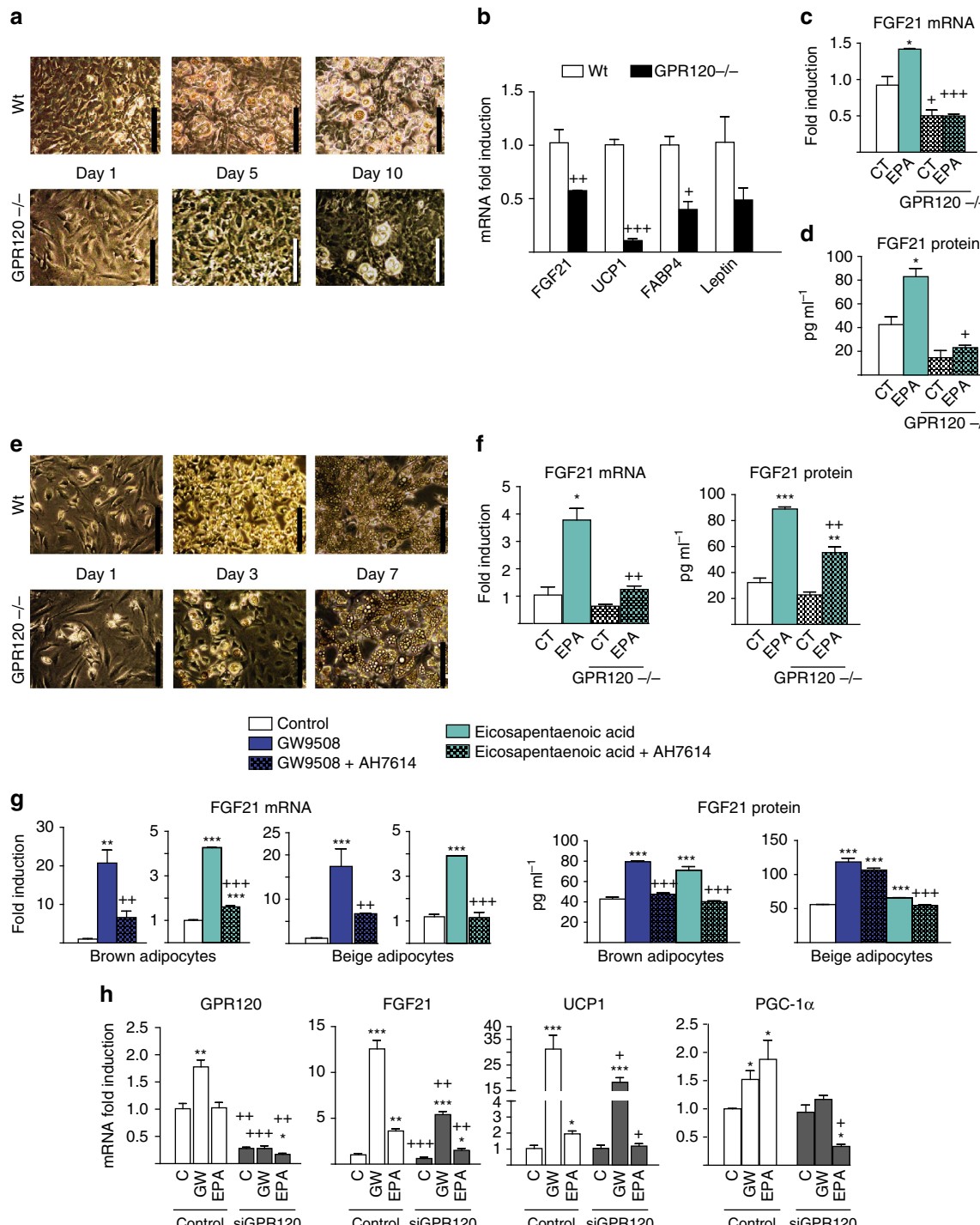

**Figure 8 | GPR120 is required for the effects of EPA on adipocytes and FGF21 induction and release.** For **a**–**d**, iBAT precursors from wild-type ($n = 3$, white) and *GPR120*-null ($n = 5$, black) mice were differentiated. (**a**) Representative optical microscopy images (scale bar, 200 μm). (**b**) Relative mRNA expression levels of *FGF21*, *UCP1*, *FABP4* and *leptin*. (**c**,**d**) Effects of EPA on *FGF21* mRNA expression and FGF21 secretion. For **e** and **f**, iWAT precursors from wild-type ($n = 3$) and *GPR120*-null ($n = 5$) mice were differentiated into beige adipocytes. (**e**) Representative optical microscopic images (scale bar, 200 μm). (**f**) Effects of EPA on *FGF21* mRNA expression and FGF21 protein secretion. (**g**) Differentiated brown and beige adipocytes were treated with GW9508 (blue bars) or EPA (turquoise bars) in the presence or absence of AH7614 (a GPR120 antagonist, patterned bars) for 24 h ($n = 3$). *FGF21* mRNA expression and FGF21 protein levels in culture medium. (**h**) Differentiated brown adipocytes were subjected to siRNA-mediated knockdown of GPR120 (see the Methods section) and treated with GW9508 or EPA. mRNA expression levels of *GPR120*, *FGF21*, *PGC-1α* and *UCP1* ($n = 3$). Bars are means + s.e.m. (\*$P < 0.05$, \*\*$P < 0.01$ and \*\*\*$P < 0.001$ relative to controls, and +$P < 0.05$, ++$P < 0.01$ and +++$P < 0.001$ for comparisons between wild-type and *GPR120*-null cells (**a**–**f**), the effects due to AH7614 (**g**), and the effects due to siRNA-GPR120 (**h**). For **b**, two-tailed unpaired Student's *t*-test was performed; for **c**–**h**, analysis of variance with Tukey's *post hoc* test).

FGF21 expression in brown adipocytes involves the activation of p38 MAPK and its downstream induction of FGF21 gene transcription.

**FGF21 is involved in the effects of GPR120 activation**. FGF21, which is a hormonal factor expressed and released by brown and beige cells[19,20], has strong autocrine/endocrine effects on BAT activation and WAT browning[36,37]. Thus, we next explored the involvement of FGF21 in the biological response to GPR120 activation.

*FGF21*-null mice were treated with GW9508 in their diet for 1 week. The absence of FGF21 strongly impaired the capacity of GW9508 to induce BAT thermogenic activation, as evidenced by decreased induction of thermogenic genes (for example, *UCP1*, *PGC-1α* and *Bmp8b*) and *Glut1*, whereas markers of general adipogenesis (for example, *FABP4*) were unaltered (Fig. 9a). *FGF21*-null mice kept under basal conditions exhibited increased lipid content, indicative of impaired activity and 'whitening' of BAT (Fig. 9b). GW9508 treatment of *FGF21*-null mice, in contrast with wild-type mice, even increased the signs of impaired thermogenic activity and 'whitening' of BAT as evidenced by increased lipid content and enhanced leptin gene expression. Moreover, GW9508 failed to increase UCP1 protein levels in BAT from *FGF21*-null mice (Fig. 9b).

*FGF21*-null mice also showed strong impairment of iWAT browning in response to GW9508. The clusters of multilocular, beige adipocytes, seen in GW9508-treated wild-type mice were largely absent in GW9508-treated *FGF21*-null mice (Fig. 9d), and there were strong reductions in the GW9508-mediated upregulation of the thermogenic, beige phenotype-related, genes (for example, *PGC-1α*, *UCP1* and *Sirt3*; Fig. 9c). GW9508 treatment increased UCP1 protein levels in iWAT from wild-type mice and the extent of induction of UCP1 protein levels in *FGF21*-null mice was significantly lower. Although the response of eWAT to GW9508-induced browning was less marked than that in iWAT of wild-type mice, the mild upregulation observed among browning-associated genes was blunted in GW9508-treated *FGF21*-null mice (Supplementary Fig 10).

We further explored the cell-autonomous functions of FGF21 in the responsiveness to GPR120 activation. Brown adipocytes from *FGF21*-null mice differentiated normally *in vitro*, and GPR120 expression was not altered by the lack of FGF21. GW9508 was able to activate the expressions of thermogenic genes (*UCP1*, *PGC-1α*, *COXIV* and *Sirt3*) in cultured brown adipocytes, but the extent of these inductions were significantly less in *FGF21*-null cells than in control cells (Fig. 10a). When previously differentiated brown adipocytes from *FGF21*-null mice were treated with GW9508 for 24 h, we observed reductions in the upregulations of some thermogenic genes (Fig. 10b) and glucose oxidation (Fig. 10d). In beige adipocytes differentiated from iWAT, the lack of FGF21 reduced the GW9508-induced upregulations of beige markers (*UCP1* and *PGC-1α*; Fig. 10c) and blocked the ability of GW9508 to induce glucose oxidation (Fig. 10d).

Collectively, these results indicate that at least some of the effects of GPR120 activation on the thermogenic activations of brown and beige adipocytes involve FGF21, potentially via autocrine/endocrine actions of FGF21 secreted downstream of GPR120 activation.

## Discussion
The identification of novel regulators that promote energy expenditure and the capacity of BAT activity to oxidize metabolic substrates may provide key therapeutic targets in obesity, diabetes and dyslipidemias. Attempts to promote energy expenditure by the use of sympathomimetics, which take advantage of the classical adrenergic pathway for controlling of BAT activity, have failed due to the important side effects[38].

We presently identified a novel pathway regulated by the fatty acid receptor GPR120 that controls BAT activity and WAT browning. Our data indicate that GPR120 activation plays a dual role, namely: (1) it promotes the differentiation of pre-adipocytes into the brown and beige lineages; and (2) it promotes thermogenic activation in differentiated brown and beige adipocytes. Previous reports showed that GPR120 activation increases the energy expenditure and ameliorates insulin resistance[13,14]. Given that BAT activity and WAT browning have major impacts on energy expenditure and glucose homoeostasis, our current data provide a biological basis and mechanistic explanation for these observations. This adds a new function for GPR120, in addition to its ability to decrease chronic inflammation, in association with the improved insulin resistance[13,39]. Modifications in the pro- and anti-inflammatory profiles of immune cells were recently reported to influence BAT activation and especially WAT browning[40,41]. Further research is warranted to determine the role of GPR120-mediated regulation of immune cells on BAT and browning of WAT *in vivo*; however, our current findings establish a direct, cell-autonomous effect, of GPR120 in the promotion of brown and beige cell differentiation and activation.

The effects of GPR120 activation on BAT activity and WAT browning appear to mediate the actions of omega-3 PUFAs, which are physiological activators of the GPR120 receptor, on these processes. Dietary enrichment with omega-3 PUFAs have beneficial effects on metabolic health in healthy lean individuals, whereas in experimental rodent models, omega-3 PUFA have been shown to prevent the development of obesity, and decrease hyperglycaemia and dyslipidemia[42]. Several reports agreed that dietary supplementation with PUFAs promote BAT recruitment and WAT browning as reflected by enhanced mitochondrial oxidative capacity and, in some reports, increased UCP1 content[43–47]. Single dietary supplementation with EPA (but not DHA) has reportedly had the most significant effects[43], and recent data show an especially active effect of EPA in the promotion of browning *in vitro*[48]. The molecular mechanisms underlying these effects remained to be clarified. Our current data indicate that omega-3 PUFAs have cell-autonomous GPR120-mediated effects directly stimulating the differentiation and activation of brown and beige adipocytes. This strongly supports the notion that GPR120 activation is involved in the actions of dietary omega-3 PUFAs on BAT activation, WAT browning and metabolic consequences, and may provide a molecular explanation for some of the beneficial effects of PUFAs.

Our study also shows that some of the effects triggered by GPR120 activation are mediated via the GPR120-induced expression and release of FGF21 in brown and beige adipose tissues. Fatty acids stimulate the expression of FGF21 in hepatic cells, which are devoid of GPR120, in a PPARα-mediated manner[32]. However, this pathway does not mediate FGF21 gene expression in BAT[19]. Our present data indicate that GPR120 activation is part of a highly tissue-specific pathway that regulates FGF21 expression via the p38 MAPK-mediated activation of *FGF21* gene transcription in brown and beige cell lineages. Moreover, the effects of GPR120 activation to promote BAT activity and WAT browning are mediated at least in part by FGF21. FGF21 promotes thermogenic activation of BAT, and the browning of WAT in association with increased glucose uptake and oxidation[36,37]. Similar effects were seen following GPR120 activation, and these effects were reduced in FGF21-inactivated animal and cell models. Thus, we herein report that autocrine and (perhaps) endocrine actions of FGF21 are relevant to the GPR120 activation-mediated induction of BAT differentiation and WAT

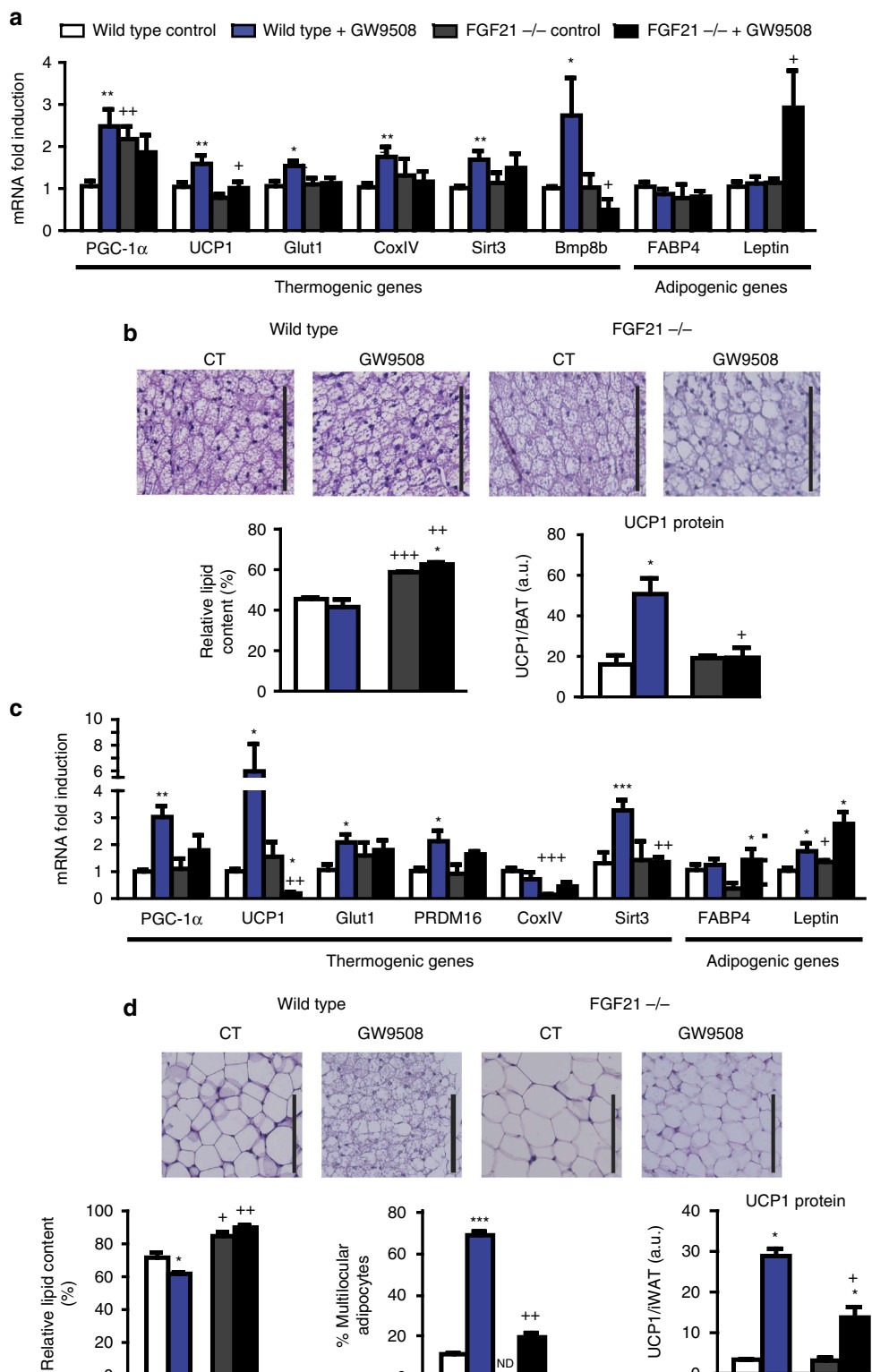

**Figure 9 | FGF21 gene invalidation reduces the effects of GW9508 treatment in mice.** Wild-type and *FGF21*-null mice were fed a control diet (white and grey bars, respectively) or supplemented with GW9508 (blue and black bars, respectively) for 7 days ($n = 5$). (**a**) Relative mRNA levels of thermogenesis-related and adipogenic genes in iBAT (**c**) and iWAT. Representative optical microscopy images of H&E staining (scale bar, 125 μm), the relative lipid content and UCP1 protein levels in iBAT (**b**) and iWAT (**d**), and the percentage of multilocular adipocytes in iWAT. Bars are means + s.e.m. (*$P < 0.05$, **$P < 0.01$ and ***$P < 0.001$ relative to untreated control mice of each genotype; +$P < 0.05$, + +$P < 0.01$ and + + +$P < 0.001$ relative to same treatment of the wild-type group; analysis of variance with Tukey's *post hoc* test).

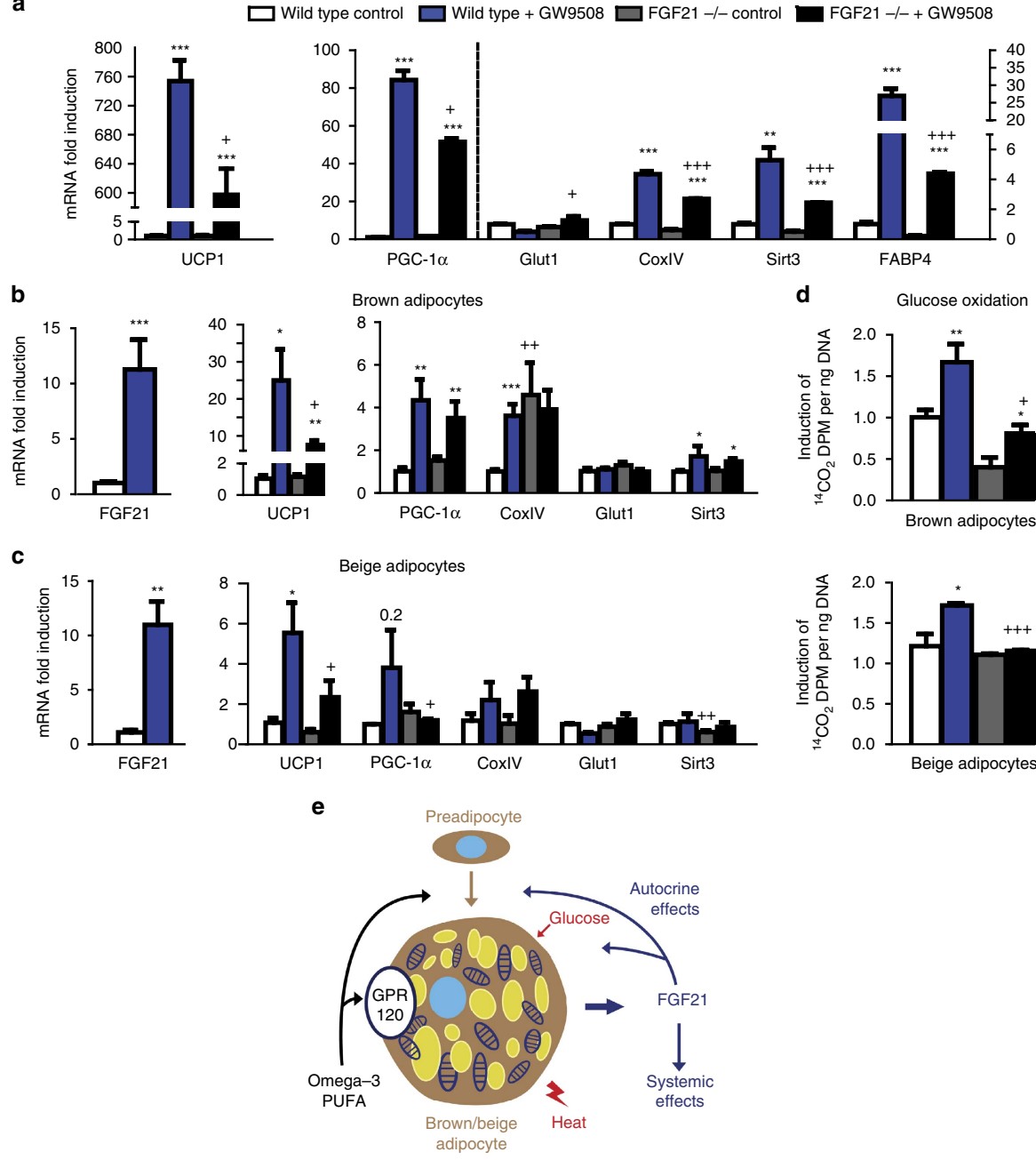

**Figure 10 | Impaired effects of GW9508 on FGF21-null brown and beige adipocytes.** (**a**) iBAT precursors from wild-type and FGF21-null mice ($n = 5$) were treated with GW9508 during differentiation, relative transcript levels of *UCP1*, *PGC-1α*, *Glut1*, *CoxIV*, *Sirt3* and *FABP4*. (**b**) iBAT precursors from wild-type and *FGF21*-null mice ($n = 5$) were differentiated and acutely treated with GW9508 (24 h), mRNA expression levels of *FGF21*, *UCP1*, *PGC-1α*, *CoxIV*, *Glut1* and *Sirt3*. (**c**) iWAT precursors from wild-type and *FGF21*-null mice ($n = 5$) were differentiated and acutely treated with GW9508 (24 h), mRNA expression levels of *FGF21*, *UCP1*, *PGC-1α*, *CoxIV*, *Glut1* and *Sirt3*. (**d**) Glucose oxidation in iBAT and iWAT-derived adipocytes from wild-type and *FGF21*-null mice after 24 h treatment with GW9508. Bars are means + s.e.m. (*$P < 0.05$, **$P < 0.01$ and ***$P < 0.001$ for the effects of GW9508; and $+ P < 0.05$, $+ + P < 0.01$ $+ + + P < 0.001$ for comparisons between wild-type and *FGF21*-null cells; and analysis of variance with Tukey's *post hoc* test). (**e**) Schematic representation of the effects of GPR120 activation by n-3 PUFAs on brown and beige adipocytes. FGF21 is involved in the GPR120 activation-mediated thermogenic activation of BAT and WAT via autocrine/endocrine mechanisms.

browning (Fig. 10e). Of note, the capacity of GPR120 activation to modify intracellular kinases (for example, p38 MAPK) suggests that GPR120 may have additional and direct effects on BAT activity and WAT browning.

In conclusion, we identified a novel pathway of BAT activation and browning of WAT based on a strong, cell autonomous, capacity of the lipid sensor GPR120 to trigger these processes. This pathway is expected to contribute to the systemic metabolic

benefits occurring after GPR120 activation. Our present findings reinforce the current interest in the potential of GPR120-activating drugs or dietary molecules for treating metabolic diseases.

## Methods

**RNA-seq and RNA-seq data analysis.** Total RNA was isolated using the RNeasy Mini Kit (Qiagen), which favours purification of RNA molecules longer than 200

nucleotides. Further sample preparation was done as described by the manufacturer (Illumina, Eindhoven, The Netherlands). Briefly, mRNA was purified from 2 μg total RNA using oligo (dT) beads, and then fragmented and randomly primed for reverse transcription followed by second-strand synthesis to create double-stranded cDNA fragments. The generated cDNA had undergone paired-end repair to convert overhangs into blunt ends. After 3′-monoadenylation and adaptor ligation, the cDNAs were purified by 2% agarose gel electrophoresis and 200-bp products were excised from the gel. Following gel digestion, the purified cDNA was amplified by PCR using primers specific for the ligated adaptors. Before sequencing, the generated libraries were submitted to a quality control assessment with an Agilent bioanalyzer 2100 (Agilent Technologies, Wokingham, UK). The RNA integrity number values for all samples were at least 7.5. To verify the cDNA quality and quantity, 1 μl cDNA was loaded on an Agilent DNA chip (DNA-1000, Agilent Technologies). Only libraries that yielded satisfactory results from the quality control studies were sequenced on an Illumina HiSeq 2,000 sequencer (DNAvision, Charleroi, Belgium) with an average of 45 million reads per sample. This level of coverage was previously shown to provide sufficient sequencing depth for the quantification of gene expression and detection of transcripts[49] and has been used in our previous studies[50,51].The obtained 100 nucleotide paired-end reads were mapped to the mouse genome (version GRCm38) using the Tophat mapper. Using this approach, we were able to map 87% of the raw reads on average. Sequencing reads were mapped to the mouse genome (*Mus musculus* version GRCm38) using Tophat version 2.02 (ref. 52) under default options for paired-end read mapping. Mapped reads were used to quantify transcripts from the Ensembl version 73 gene annotation data set (http://www.ensembl.org) using the Flux Capacitor approach. This strategy deconvolutes reads mapping to exonic regions shared by multiple transcripts by optimizing a system of linear equations, and thus specifically assigns some reads to each alternative splice form (http://flux.sammeth.net)[53]. All genes and transcripts were assigned relative coverage rates as measured in RPKM units ('reads per kilobase per million mapped reads')[54]. Lists of differentially expressed genes and transcripts were generated from the Flux Capacitor output using scripts in Perl or R. The 88,346 transcripts annotated in Ensembl 73 corresponded to 33,358 genes. Mice kept under the thermoneutral temperature were found to express a median of 18,494 genes. Of them, 15,971 genes were found in all tested individuals.

To define the genes that were upregulated or downregulated by cold, the fold change was calculated as the proportion between the sum of the RPKM for all gene transcripts under the cold condition and the same sum in control condition. Significance was tested using a Fisher's exact test (the number of reads mapped to a given gene and number of reads mapped to all other genes in the cold condition versus the control condition) and corrected by the Benjamini–Hochberg method (taking for each gene the four samples as independent tests). A difference in gene expression was considered significant if the corrected $P$ value was <0.05. As additional criteria, a gene was considered to be 'modified by cold' only if its expression changed significantly in the same direction—that is, 'up' or 'down'—in at least three out of four samples per group and no significant change in the opposite direction was observed, as in previous studies[50,51].

**Animals and treatments *in vivo*.** For RNA-seq analysis, eight adult (5 months old at the moment of the experiment) male C57BL6 mice were obtained from Harlan Laboratories and were maintained at thermoneutral temperature (29 °C). After 2 weeks, four mice randomly were placed at an environment temperature of 4 °C for 24 h. Cold-exposed mice and thermoneutral controls were killed by decapitation, iBAT was dissected and frozen for further analysis. *Fgf21*$^{-/-}$ mice (B6N;129S5-*Fgf21*$^{tm1Lex}$/Mmcd) were obtained from the Mutant Mouse Regional Resource Center (MMRRC), an NCRR-NIH-funded strain repository, having been donated to the MMRRC by Genentech, Inc. *Pparα*-null mice (B6.129S4-*Ppara*$^{tm1Gonz}$/J) were obtained from Jackson Laboratory (Bar Harbor, ME, USA). *GPR120*−/− mice (*Ffar4*$^{tm1(KOMP)Vlcg}$) were purchased from MMRRC. Adult (5 months old) male wild-type littermates were used as controls for all experiments with *Fgf21*-null mice, *Pparα*-null mice and *GPR120*-null mice. When indicated, mice were injected intraperitoneal with 1 mg kg$^{-1}$ CL316,243.

GW9508 (Cayman Chemical) was administered to adult (5 months old) male C57BL6 mice through the diet at a dose calculated to yield an intake of 50 μg g$^{-1}$ body weight per day GW9508 intake[55] for 1 week. The diet was prepared as previously described[56] by soaking diet pellets in an acetone solution of the drug; the control diet in was also soaked in acetone and dried. For cold-exposure treatments, *GPR120*-null, *FGF21*-null and the corresponding wild-type littermates were exposed to 4 °C ambient temperature during the indicated timing. Mice were subjected to non-invasive measurements (see below), and then they were killed, and blood and tissues were collected. Tissue samples were frozen for further RNA and/or protein analysis, or placed in 10% buffered formalin overnight and processed for hematoxylin and eosin (H&E) staining and optical microscopy using standard procedures. The relative lipid content and percentage of multilocular adipocytes were measured from optical microscopy images using ImageJ and CellProfiler software, with at least five independent preparations quantified for each experimental group. Data of the relative lipid content and the area occupied by multilocular adipocytes were expressed as percentages relative to total image area.

The volume of consumed oxygen, the volume of produced carbon dioxide and the respiratory quotient were determined (Harvard Apparatus). Rectal temperature

was determined using an electronic thermistor equipped with a rectal probe. Body temperature was non-invasively estimated by measuring the eye-surface temperature using a high-sensitivity infrared thermography camera (FLIR T335), as previously reported[57,58].

All experiments were performed in accordance with European Community Council directive 86/609/EEC and experiments, as well as the number of animals to be used were approved by the Institutional Animal Care and Use Committee at the University of Barcelona based on the expected effects size.

**Serum biochemistry.** Glucose and triglyceride levels were measured using Accutrend Technology (Roche Diagnostics, Basel, Switzerland). FGF21 levels were quantified with ELISA (RD291108200R, Biovendor). Insulin, adipokines and cytokines were quantified using a Multiplex system (MADKMAG-HK, Millipore).

**Brown and beige adipocyte cell cultures.** Stromal vascular cells were obtained from iBAT and iWAT excised from 3-week-old C57BL6 mice (males and females), primary cultures were generated and the cells were induced to differentiate into brown and beige adipocytes, respectively, following the previously reported procedures[25,26]. Brown adipocyte differentiation was achieved by exposing confluent precursor cells from iBAT in DMEM/F12 medium containing 10% foetal bovine serum (FBS) and supplemented with 20 nM insulin, 2 nM T3 and 0.1 mM ascorbic acid (ITA).

For beige cell differentiation, confluent precursor cells from iWAT and eWAT were maintained in DMEM/F12 containing 10% newborn calf serum (NCS). For differentiarion, 850 nM insulin, 3 μM T3, 35 nM dexamethasone and 10 μM rosiglitazone were added. When indicated, pre-adipocytes were cultured in the presence of delipidated serum (Charcoal Stripped Serum-GIBCO) instead of FBS/NCS

Cells were treated either across the differentiation process or were treated acutely (24 h), when already differentiated. Treatments included TUG-891 (200 μM), grifolic acid (100 μM), GW9508 (100 μM), ALA (100 μM), EPA (100 μM), NE (0.5 μM), dibutyryl-cAMP (1 mM), SB202190 (10 μM), H89 (20 μM), GW7647 (1 μM), GW9662 (30 μM), AH-7614 (100 μM), U-0128 (10 μM), compound C (10 μM), wortmannin (2 μM) and CL316,243 (1 μM). All reagents were obtained from Sigma with the exception of TUG-891, AH-7614, GW9662 (from Tocris), GW9508 (from Cayman Chemical), compound C (Calbiochem) and U-0128 (Enzo). When indicated, cells were subjected to dynamic measurements (see below) and/or further collected for RNA extraction.

**Dynamic measurements of adipocytes in culture.** Glucose oxidation rates were determined in cultured adipocytes. Cells were incubated for 60 min in the appropriate cell culture medium containing [$^{14}$C]-glucose after which trapped $^{14}$C-CO$_2$ was measured. Cellular heat production was measured by infrared thermography in accordance with an initial report[59] further developed for application to brown adipocytes[27,28]. Basically, brown or beige adipocytes were grown in a 12-well plate and placed on a 37 °C heat block in a polystyrene box coated with black paper to optimize insulation. Images were acquired by an infrared camera (FLIR systems T335, Wilsonville, OR, USA), which detects a 7.5–13 μm spectral response with a thermal sensitivity of 0.1 °C, and analysed using the FLIR Quick Report software (Wilsonville, OR, USA). Supplementary Figure 11 shows an example of thermographic recording of brown adipocytes in culture. Oxygen consumption of cells was recorded using the Oxygen Biosensor System (BD) in the absence or presence (uncoupled respiration) of 10 μg ml$^{-1}$ oligomycin, and also after exposure of cells to CL316,243 for 24 h, as previoulsy described[36].

**siRNA-mediated interference.** Reverse transfection of differentiated adipocytes was performed using the Lipofectamine RNAiMAX (InvitroGene) and Optimem (Life Technologies) reagents, random duplexes or two independent siRNA duplexes designed to silence GPR120 (67928017 and 67928014, final concentration 10 μM), all from Integrated DNA Technologies-TriFECTA. One day after transfection, the cells were exposed to various treatments and then collected for analysis of extracted RNA and conditioned media.

**Transient transfection of promoter constructs.** Reverse transfection of differentiated brown adipocytes with the plasmid −1497-FGF21-Luc, containing the 5′ region of the mouse FGF21 gene linked to the luciferase reporter gene, and the deleted form 69-FGF21-Luc[19] was performed using Lipofectamine. Where indicated, an expression plasmid for a dominant-negative form of MKK6 (MKK6-K82A, Addgene plasmid 13,519) was co-transfected (0.06 μg per well). The pRL-CMV expression plasmid for the sea pansy (Renilla reniformis) luciferase was used as an internal transfection control (Promega, Madison, WI, USA). Cells were incubated for 48 h after transfection, and where indicated, they were treated with 100 μM GW9508 or 100 μM EPA for 24 h before collecting. Firefly luciferase activity elicited by FGF21 promoter constructs was normalized for variation in transfection efficiency using Renilla luciferase as an internal standard, all measured in a Turner Designs luminometer (TD 20/20) using the Dual-Luciferase Reporter Assay system (Promega).

**Quantitative PCR with reverse transcription.** Reverse transcription was performed, using random hexamers primers (Applied Biosystems, Foster City, CA, USA)

and 0.5 µg RNA in a total reaction volume of 20 µl. For PCR, Taqman Gene Expression Assay probes were used (Supplementary Table 3), with reaction mixtures containing 1 µl cDNA, 10 µl TaqMan Universal PCR Master Mix (Applied Biosystems), 250 nM probes and 900 nM of primers from the Assays-on-Demand Gene Expression Assay Mix. The 18S rRNA was measured as the housekeeping reference gene. The mRNA level of the gene of interest in each sample was normalized to that of the reference control using the comparative $(2 - \Delta CT)$ method, according to the manufacturer's instructions. A transcript was considered to be non-detectable when $CT \geq 40$.

**Immunoblots and multiplex protein assays.** Western blot analysis of tissue and cell culture extracts was performed following standard procedures, using primary anti-UCP1 (1:1,000 ab10983, Abcam, Cambridge, UK) and anti-GPR120 (1:150sc-99105, Santa Cruz, USA). Specificity of GPR120 detection was checked using BAT extracts from male C57BL6 GPR120-null mice (Supplementary Fig. 12). Loading controls were established using α-tubulin immunoblots (T9026, Sigma-Aldrich) or Ponceau staining of membranes. Immunoreactive proteins were detected using an ECL (enhanced chemiluminescence) system (GE Healthcare). Signal intensities were quantified by scanning densitometry (Phoretics 1D Software). Uncropped scans of western blots and corresponding Ponceau-stained membranes, including scale markers are shown in Supplementary Fig. 13. Quantitative measurements of changes in phosphorylated ERK1/2, CREB and p38 MAPK were performed using multiplex-based technology (48–680MAG and 48–681MAG kits, Millipore, Billerica, MA, USA) by determining the ratio of the corresponding phosphoproteins relative to total specific proteins.

**Statistics.** Two-tailed unpaired Student's $t$-test or one-way analysis of variance were used to test for the statistical significance of differences between two experimental conditions. Welch's correction was applied when unequal variances were detected by $F$-test, using the GraphPad statistical software (GraphPad Software Inc., San Diego, CA, USA). Statistical significance was set with an α-value of $P < 0.05$, and underlying assumptions for validity of all tests were assessed. Data are shown as means ± s.e.m.

**Data availability.** The raw data generated during the RNA-seq procedure is deposited in Gene Expression Omnibus (GEO) under accession number GSE77534. The complete list of cold-modulated genes in BAT after RNA-seq data analysis is available at http://lmedex.ulb.ac.be/data.php.

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

# ARTICLE

45. Takahashi, Y. & Ide, T. Dietary omega fatty acids affect mRNA level of brown adipose tissue uncoupling protein 1, and white adipose tissue leptin and glucose transporter 4 in the rat. *Br. J. Nutr.* **84,** 175–259 (2000).

46. Flachs, P. *et al.* Synergistic induction of lipid catabolism and anti-inflammatory lipids in white at of dietary obese mice in response to calorie restriction and omega fatty acids. *Diabetologia* **54,** 2626–2664 (2011).

47. Villarroya, J. *et al.* Fibroblast growth factor-21 and the beneficial effects of long-chain omega polyunsaturated fatty acids. *Lipids* **49,** 1081–1090 (2014).

48. Fleckenstein-Elsen, M. *et al.* Eicosapentaenoic acid and arachidonic acid differentially regulate adipogenesis, acquisition of a brite phenotype and mitochondrial function in primary human adipocytes. *Mol. Nutr. Food Res.* **60,** 2065–2075 (2016).

49. Wang, Z., Gerstein, M. & Snyder, M. RNA-Seq: a revolutionary tool for transcriptomics. *Nat. Rev. Genet.* **10,** 57–63 (2009).

50. Eizirik, D. L. *et al.* The human pancreatic islet transcriptome: expression of candidate genes for type 1 diabetes and the impact of pro-inflammatory cytokines. *PLoS Genet.* **8,** e1002552 (2012).

51. Cnop, M. *et al.* RNA sequencing identifies dysregulation of the human pancreatic islet transcriptome by the saturated fatty acid palmitate. *Diabetes* **63,** 1978–1993 (2014).

52. Trapnell, C. *et al.* Transcript assembly and quantification by RNA-Seq reveals unannotated transcripts and isoform switching during cell differentiation. *Nat. Biotechnol.* **28,** 511–516 (2010).

53. Montgomery, S. B. *et al.* Transcriptome genetics using second generation sequencing in a Caucasian population. *Nature* **464,** 773–780 (2010).

54. Mortazavi, A., Williams, B. A., McCue, K., Schaeffer, L. & Wold, B. Mapping and quantifying mammalian transcriptomes by RNA-Seq. *Nat. Methods* **5,** 621–628 (2008).

55. Ou, H. Y. *et al.* Multiple mechanisms of GW-9508, a selective G protein-coupled receptor 40 agonist, in the regulation of glucose homeostasis and insulin sensitivity. *Am. J. Physiol. Endocrinol. Metab.* **304,** E668–E676 (2013).

56. Cabrero, A. *et al.* Bezafibrate reduces mRNA levels of adipocyte markers and increases fatty acid oxidation in primary culture of adipocytes. *Diabetes* **50,** 1883–1890 (2001).

57. Johnson, S., Rao, S., Hussey, S. B., Morley, P. S. & Traub-Dargatz, J. L. Thermographic eye temperature as an index to body temperature in ponies. *J. Equine. Vet. Sci.* **31,** 63–66 (2011).

58. Purslow, C. & Wolffsohn, J. S. The relation between physical properties of the anterior eye and ocular surface temperature. *Optom. Vis. Sci.* **84,** 197–201 (2007).

59. Paulik, M. A. *et al.* Development of infrared imaging to measure thermogenesis in cell culture: thermogenic effects of uncoupling protein-2, troglitazone, and beta-adrenoceptor agonists. *Pharm. Res.* **15,** 944–949 (1998).

## Acknowledgements

We thank A. Peró and M. Morales for technical support. Scientific and technical advice by Dr C. Wolfrum and M. Christian is acknowledged. This work has been supported by Grants SAF2014-55725 and PI14/00063 from the Ministerio de Economia y Competitividad (MINECO), Spain, and co-financed by the European Regional Development Fund (ERDF), and by the European Community's Seventh Framework Program (FP7 BetaBat for F.V. and D.L.E.) and the Horizon 2020 Program (T2Dsystems (GA667191)) to D.L.E. T.Q.-L. was supported by a CONACyT (National Council for Science and Technology in Mexico) Ph. D. scholarship. R.M. and M.C were supported by PhD scholarships from MINECO, Spain.

## Authors contributions

The experiments were conceived and designed by T.Q.-L., D.L.E. and F.V.; RNA-seq data were obtained and analysed by J.-V.T., R.C. and D.L.E.; experiments with mice were performed by T.Q.-L., R.M., A.P. and R.I.; cell culture experiments were performed by T.Q.-L., M.G., M.P. and A.G.-N., analysis of microscopy images was performed by M.C. and M.G.; overall data were analysed by T.Q.-L., M.G., D.L.E. and F.V. The paper was written by F.V. and revised/approved by all contributors.

## Additional information

**Competing financial interests:** The authors declare no competing financial interests.

