## [Peer Review File · Nature Communications]

Reviewers' comments:

Reviewer #1 (Remarks to the Author):

This is a very nice study describing a brown fat cell autonomous role for GPR120 in mediating thermogenic function. Animal and cell culture models are used to show that synthetic and natural GPR120 agonists promote classic BAT activation and a browning response in white fat. Interestingly, GPR120 activation robustly triggers FGF21 production and this is required (at least partly) for BAT activation. In general, I think this is a strong paper which will be of broad interest. However, functional studies showing the effect of GPR120 agonism on thermogenesis are lacking and should be performed. Specific comments follow:

1. There is no functional assessment of brown fat thermogenesis in mice (or cells) following treatment with GPR120 agonists or in GPR120 knockout mice. This can be nicely evaluated in mice by measuring oxygen consumption in response to beta-agonists. Similarly, thermogenesis is routinely measured in cell culture models through assaying oxygen consumption before and after stimulation with beta agonists. Given the premise that GPR120-signaling increases BAT thermogenesis, this really needs to be determined experimentally.

The temperature measurements in cell cultures used here (fig 5h, 6h etc.) are not adequately described and are almost certainly not reflective of UCP1-action given that the cells are not stimulated by b-agonists.

2. UCP1 western blots should be provided for data in Figs 3a, b and 4c.

3. Fig 9(a). It is interesting to note that UCP1 is not reduced in the FGF21 $-/-$ brown fat. Does this imply that GPR120 activation (and consequential FGF21 production) is not part of the physiological process of brown fat activation?

4. Fig 10a. The FGF21 requirement for GW9508 effects are very modest in the cell culture system. The authors should comment here.

5. Fig 7a-d. Check figures- the legend is not compatible with all of the graphs.

6. A control for H89 treatment in 1f is needed- did the drug treatment work as intended?

7. Have the authors treated GPR120 $-/-$ mice with GW9508 to verify that the effects of the drug are mediated by GPR120 signaling.

8. Some of the figures should be condensed and/or moved to supplement. There are too many redundant results in the Figures, which make the paper harder to wade through. For example, Figure 1 (and much of the accompanying description in the results) are unnecessary and uninformative. The pathway analysis does not provide any important insight that pertain to the paper. Just start results by saying GPR120 was one of the highest induced genes by cold (p6, Fig 2). Fig 5-6; 7-8 and 9-10 could be consolidated. In my view, this paper should have no more than 6 main figures.

Reviewer #2 (Remarks to the Author):

The study is interesting and has some novelty, revealing a pathway operating in GPR120-mediated BAT activation and browning of WAT with FGF21 involvement. However, the molecular basis

(mechanism) of the physiological phenotype between GPR120 and FGF21 is poorly characterized.

Major points:

1) In Figure 2a, why did authors use small intestine to compare the different tissues? Show Q-Values. Usually people get these high BAT values when their housekeeping gene is expressed very low.

2) In Figure 2b, why was BAT GPR120 expression reduced after 21 days? Similar question as for BAT, why was GPR120 increased after 24hrs (short-term cold exposure) and decreased afterward? Could authors suggest possible explanation for the effects of short- and long-term cold exposure on GPR120 expression level in BAT vs. iWAT?

3) While GPR120 is activated by omega-3 fatty acids (e.g. ALA, EPA, and DHA), authors have used these all three omega-3 fatty acids, but claimed DHA had barely significant effects on BAT differentiation. Authors have also mentioned in the discussion that Outart et al (51) has shown better effects with EPA on browning than DHA. In Figure 8f and 8g, EPA-induced expression of FGF21 is reduced in GPR120 KO mice and GPR120 antagonist treatment, respectively, but FGF21 level is still increased compared to the controls. So it seems that EPA does have GPR120-independent effects on FGF21 induction in iWAT and brown adipocytes. Authors have to explain or at least discuss that possible different activities among the different omega-3 fatty acid species and GPR120-independent effect on FGF21 induction by EPA.

4) Page 16-17 at the end of result description, authors have mentioned that GPR120 activates FGF21 and then leads to browning. However, it's not only FGF21, as browning/beiging through activation of GPR120 in FGF21 KO mice is still present, just a bit reduced (Figure 10). So it clearly implies that there are other pathways. What can be the other pathways that GPR120-mediated browning/beiging beside of FGF21. Since FGF21 is not the only one factor for GPR120-mediated browning/beiging effects, authors should suggest or discuss this point.

5) Overall, how does GPR120 activation induce FGF21 level to regulate browning and beiging phenotype? The authors fail to show the mechanism how this might happen. All the data in this study is quite descriptive and repetitive with different depots of adipose tissue without direct molecular mechanism.

6) Although authors claimed that the negligible expression of GPR40 in BAT would justify their using of GW9508 in the set of studies, using GW9508 in FGF21 KO mice (Figure 9) is not appropriate model in the context of authors' study which they want to emphasize the link between GPR120 and FGF21. GW9508 in animal model will hit GPR40 with stronger activity (for EC50, ~47nM for GPR40 vs ~2.2 uM for GPR120) than to activate GPR120. Although authors described the absence of FGF21 impaired the capacity of GW9508 (in other words GPR120 activation, in the authors' point of view), this data is not clear enough to draw line between GPR120 and FGF21. Authors should use GPR120 specific agonist (TUG891 or compound A from the recent GPR120 agonist publications), not GW9508 for in vivo study, unless authors generate GPR120 and FGF21 double KO mice to prove the link between GPR120 and FGF21.

7) Although the (online) methods are very detailed and informative, the legends of the main figures could use additional description so that the reader does not have to refer back and forth between sections. Authors also need to carefully label the subset of figures with their description in each figures legend. For example, figure 8 legend is very difficult to figure out what to what. If the legend can be modified like this, it will be much understandable without again "refer back and forth between sections".

Example for modification of figure 8 legend; For a-d, iBAT precursors from wild-type and GPR120-null

mice were differentiated (n=3-5). (a) Representative optical microscopic images (magnification 10X). Relative mRNA expression levels of FGF21, UCP1, FABP4 and leptin (b). Effects of EPA on FGF21 mRNA expression and FGF21 secretion (c and d). For e and f, iWAT precursors from wild-type and GPR120-null mice were differentiated into beige adipocytes (n=3-5). Representative optical microscopic images (magnification 10X) (e). Effects of EPA on FGF21 mRNA expression and FGF21 protein secretion (f). (g) Differentiated brown and beige adipocytes were treated with GW9508 or EPA in the presence or absence of AH7614 (a GPR120 antagonist) for 24 h (n=3). FGF21 mRNA expression and FGF21 protein levels in culture medium. (h) Differentiated brown adipocytes were subjected to siRNA-mediated knockdown of GPR120 (see Methods section) and treated with GW9508 or EPA. mRNA expression levels of GPR120, FGF21 and UCP1 (n=3).

Minor points:

- 1) In Figure 3, 3a shows Leptin, 3b shows AdipoQ in adipogenic genes and Bmp8b is pretty much the highest increased with treatment in 3a, but not in 3b? Looks a bit like pick and choose.
- 2) In Figure 7c and d, where is ALA? What about FGF21m RNA expression? In 7g, where is dilapidated condition in FGF21 mRNA graph and ALA in FGF21 protein graph? Again, looks a bit like pick and choose, not consistent, especially when authors show the "same" graphs within one figure.
- 3) Authors need to check their citations thoroughly. The reference number they cited and actual content they would describe are not matched. For example, line 7 from bottom of page 10, the reference they cited is 31, should be 13 for GPR120 study.

Reviewer #3 (Remarks to the Author):

The current manuscript provides the molecular mechanisms by which GPR120 controls brown fat thermogenesis through activating FGF21 expression. The authors provided ample data in mice and cultured cells that reinforce the importance of GPR120 in whole body energy homeostasis by promoting brown and beige fat development. This would be an important addition for the future therapeutic intervention using GPR agonists. Overall the data support the authors' conclusions --- the critical experiment missing in the current manuscript is to show the effect of GW9508 in GPR120 KO mice. This would be crucial to establish the causal link between GW9508 and brown and beige fat in vivo.

1. As described above, it would be critical to examine the extent to which the GW9508 effects on BAT thermogenesis and browning are affected in GPR120 KO mice in vivo.
2. Related to this point, the authors should examine if GW9508 increases whole body energy expenditure in wt and GPR120 KO mice.
3. The effects of GW in FGF21 KO mice are intriguing. Now the authors have an opportunity to examine how much of the metabolic improving effects of GW9508, such as improved insulin resistance, are blunted in FGF21 KO mice under diet induced obesity. This would significantly increase the impact of this paper.

Minor points:

1. The authors should provide GPR120 protein expression in key parts of Fig.2.
2. The authors wish to provide UCP1 protein data in some of the critical figures like Fig.9 and 10.
3. While liver is the major source of serum FGF21, pancreas can contribute to serum FGF21 level. It would be helpful to test if GW9508 regulates FGF21 expression in the pancreas.

Reviewer #1 (Remarks to the Author):

This is a very nice study describing a brown fat cell autonomous role for GPR120 in mediating thermogenic function. Animal and cell culture models are used to show that synthetic and natural GPR120 agonists promote classic BAT activation and a browning response in white fat. Interestingly, GPR120 activation robustly triggers FGF21 production and this is required (at least partly) for BAT activation. In general, I think this is a strong paper which will be of broad interest. However, functional studies showing the effect of GPR120 agonism on thermogenesis are lacking and should be performed. Specific comments follow:

We thank the reviewer for these positive comments on our study.

1. There is no functional assessment of brown fat thermogenesis in mice (or cells) following treatment with GPR120 agonists or in GPR120 knockout mice. This can be nicely evaluated in mice by measuring oxygen consumption in response to beta-agonists. Similarly, thermogenesis is routinely measured in cell culture models through assaying oxygen consumption before and after stimulation with beta agonists. Given the premise that GPR120-signaling increases BAT thermogenesis, this really needs to be determined experimentally.

To address this relevant point, we measured oxygen consumption in mice under basal conditions and in response to the β 3-agonist CL316,243 in mice, as shown in Figure 3a of the revised manuscript. Consistent with our findings that BAT recruitment and iWAT browning were induced by the GPR120 agonist GW9508, these new data indicated that GW9508 increased oxygen consumption and enhanced the response to CL316,243 in wild-type mice. GPR120-null mice showed a mild reduction in oxygen consumption relative to wild-type mice; this was enhanced following GW9508 treatment, especially after CL316,243 injection.

We have also added data on oxygen consumption in cultured cells treated with GW9508 or EPA, with and without exposure to CL316,243. Our results indicate that GW9508 increases basal, oligomycin-resistant and (to an even greater degree) CL316,243-induced respiration in brown adipocytes (see Fig 5e in the revised manuscript). Treatment of cells with EPA yielded results that were similar, but somewhat less marked (see Fig 7c). These findings are in line with the original in vitro data on ¹⁴C-glucose oxidation, and confirm the functional and cell-autonomous effects of GPR120 activation in relation to BAT recruitment and iWAT browning.

The temperature measurements in cell cultures used here (fig 5h, 6h etc.) are not adequately described and are almost certainly not reflective of UCP1-action given that the cells are not stimulated by b-agonists.

We agree that assays of oxygen consumption and glucose oxidation are more adequate and validated choices for assessing the oxidative activity in cells (see above). However, when we assessed heat production in brown and beige cells by high sensitive thermography using the

procedure described by Celi's group in the Cell Metab (2014,19,302) and Int J Obesity (2014,38,170) papers, the generated data were reproducible and consistent with our oxygen consumption and glucose oxidation data. As reported by the Celi's group in response to factors such as FDNC5, we found some effects of GPR120 agonists even in the absence of β -agonists, likely reflecting overall enhancement of oxidative capacity within cells. As these assays involved a methodology that is totally independent of other measures of oxidative activity, we believe that the obtained data are worthy of being included in our manuscript. However, these observations can be removed if requested by the editor. In the revised manuscript, we have expanded the methodological details regarding this assay (Supplemental section) to provide additional clarity.

2. UCP1 western blots should be provided for data in Figs 3a, b and 4c.

Western blot-based quantifications of UCP1 protein are now provided in Fig 2, Fig 3 and Fig 9 in the revised manuscript.

3. Fig 9(a). It is interesting to note that UCP1 is not reduced in the FGF21 $-/-$ brown fat. Does this imply that GPR120 -activation (and consequential FGF21 production) is not part of the physiological process of brown fat activation?

Effectively, despite the marked impairment in the UCP1 mRNA response to GW9508 in FGF21-KO mice, there are no effects of FGF21 invalidation on basal expression of UCP1 in BAT and iWAT; this has been observed previously by our group and others (De Sousa-Coelho et al. J Lipid Res, 2013; 54, 1786) . Under basal conditions mice are fed a standard chow, high in carbohydrates and low in fat, and are not exposed to a strong thermal challenge for BAT activation. If the GPR120/FGF21 pathway represent a putative mechanism involved in the n-3 PUFA-dependent regulation of BAT, as our data suggest, this pathway may be not sufficiently stimulated under those basal conditions.

4. Fig 10a. The FGF21 requirement for GW9508 effects are very modest in the cell culture system. The authors should comment here.

It is true that, although the changes on transcript levels in this in vitro experimental setting were significant, they were not dramatic, especially in differentiating brown adipocytes. However, the impression that these effects are very modest was probably enhanced by the "cut" scale that we used in certain panels (e.g. for the UCP1 mRNA). We have amended the utilized scale in the revised manuscript. In fact, the GW9508-triggered induction of UCP1 mRNA in FGF21-KO differentiated brown and beige adipocytes (panels 10 b and 10 c) was somewhat less than 50% of the effect elicited in wild-type cells. Moreover, the induction of glucose oxidation by GW9508 was markedly blunted in FGF21-null cells.

We have anyway toned down our description of the results presented in Figure 10a. In addition, the Discussion section now notes that the action of GPR120 on brown adipocytes appears to have both, FGF21-dependent and FGF21-independent components (see also our reply to point 3, Reviewer 2).

5. Fig 7a-d. Check figures- the legend is not compatible with all of the graphs.

Thanks for pointing this issue. In response to this comment (and a suggestion by Reviewer 2) we have carefully checked and corrected the Figure legends.

6. A control for H89 treatment in 1f is needed- did the drug treatment work as intended?

In the revised manuscript (Supplemental Fig 1) we now show the mRNA levels of UCP1, which confirm that H89 effectively inhibited PKA in that experimental setting. UCP1 is a bona-fide target of both PKA and p38 MAPK (Frederiksson et al. Biochim Biophys Acta. 2001,1538,206; Cao et al. J Biol Chem. 2001; 276, 27077).

7. Have the authors treated GPR120 +/- mice with GW9508 to verify that the effects of the drug are mediated by GRP120 signaling.

We performed the requested experiment (which was also suggested by Reviewer 3) and now provide the relevant data in Fig 3 of the revised manuscript. We found that GPR120-null mice show impaired oxygen consumption after treatment with GW9508, especially under conditions of induction by CL316,243 (as mentioned in the reply to point 1 of this Reviewer). Importantly, GPR120-null mice had strong impairment of the GW9508-triggered inductions of circulating FGF21 levels and FGF21 expression in BAT and iWAT, and in the overall browning of WAT. Overall, these data confirm that GPR120 is critically involved in the effects elicited by GW9508.

8. Some of the figures should be condensed and/or moved to supplement. There are too many redundant results in the Figures, which make the paper harder to wade through. For example, Figure 1 (and much of the accompanying description in the results) are unnecessary and uninformative. The pathway analysis does not provide any important insight that pertain to the paper. Just start results by saying GPR120 was one of the highest induced genes by cold (p6, Fig 2). Fig 5-6; 7-8 and 9-10 could be consolidated. In my view, this paper should have no more than 6 main figures.

In accordance with the Reviewer's suggestion, we have deleted our description of the overall pathways and restricted our conclusions of RNAseq data analysis to our novel identification of GPR120 as one of the most highly cold-induced genes by cold examined to date. As a consequence the original Figure 1, with the pathway analysis was substituted by a the new Figure 1 showing the regulation of the GPR120 gene expression in brown adipose tissue and brown adipocytes.

We also moved some of the data obtained in mature, differentiated, brown and beige adipocytes to the Supplemental material. However, given the need to include new experiments and novel data requested by the various reviewers, we were not able to reach the suggested number of 6 figures.

Reviewer #2 (Remarks to the Author):

The study is interesting and has some novelty, revealing a pathway operating in GPR120-mediated BAT activation and browning of WAT with FGF21 involvement. However, the molecular basis (mechanism) of the physiological phenotype between GPR120 and FGF21 is poorly characterized.

We thank the Reviewer for these positive comments on our study. Further data on the molecular basis of the GPR120/FGF21 system in BAT have been added to the revised manuscript to better clarify the molecular basis of the observed physiological phenotypes.

Major points:

- 1) In Figure 2a, why did authors use small intestine to compare the different tissues? Show Q-Values. Usually people get these high BAT values when their housekeeping gene is expressed very low.

We used the small intestine as an example of well-established, functionally relevant, expression of GPR120 (Hirasawa et al. 2005, Nature Medicine 11, 90). In the revised manuscript, we now include data obtained from the colon, which is another established site of functional GPR120 expression (Gotho et al. Biochem Biophys Res Comm 2007, 354, 591). The high expression of GPR120 mRNA in BAT relative to other tissues was still seen when we used a reference housekeeping gene (cyclophilin) other than that used in Figure 1a (the 18s rRNA). (Supplemental Fig 1). Under the standard TaqMan qRT-PCR conditions and using 1 µg RNA, the CTs obtained for GPR120 were ~ 26 in BAT, ~ 33 in colon and ~ 39 in liver. The statistically significant differences obtained versus the small intestine are shown in Figure 1 and Supplemental Figure 1.

- 2) In Figure 2b, why was BAT GPR120 expression reduced after 21 days? Similar question as for BAT, why was GPR120 increased after 24hrs (short-term cold exposure) and decreased afterward? Could authors suggest possible explanation for the effects of short- and long-term cold exposure on GPR120 expression level in BAT vs. iWAT?

It is not unexpected to observe a more moderate relative induction of the GPR120 mRNA under long-term cold exposure of BAT and iWAT as compared to the strong induction usually observed under short-term cold exposure. This is consistent with the known behavior of other gene transcripts encoding thermogenic components, such as UCP1 and PGC-1α. The transcripts encoding these components show distinct time-courses of expression in response to cold in BAT and WAT (see, for instance, Coulter et al. Physiol Genom 2003, 14, 139) and usually exhibit their highest induction under short-term cold exposure. Long-term thermogenic induction causes overall recruitment processes in BAT and tissue remodeling in iWAT. Thus, the relative transcript levels of relevant thermogenic actors may appear to be only moderately increased relative to the constitutive RNAs used for reference, which may be increased due to tissue remodeling. These issues were previously discussed for UCP1 by Nedergaard et al. (Biochim Biophys Acta 2013, 1831, 943). We also note that, as expected, the relative protein levels of

GPR120 in BAT and iWAT exhibit a delayed time-course of response to cold induction compared with GPR120 mRNA.

- 3) While GPR120 is activated by omega-3 fatty acids (e.g. ALA, EPA, and DHA), authors have used these all three omega-3 fatty acids, but claimed DHA had barely significant effects on BAT differentiation. Authors have also mentioned in the discussion that Outart et al (51) has shown better effects with EPA on browning than DHA. In Figure 8f and 8g, EPA-induced expression of FGF21 is reduced in GPR120 KO mice and GPR120 antagonist treatment, respectively, but FGF21 level is still increased compared to the controls. So it seems that EPA does have GPR120-independent effects on FGF21 induction in iWAT and brown adipocytes. Authors have to explain or at least discuss that possible different activities among the different omega-3 fatty acid species and GPR120-independent effect on FGF21 induction by EPA.

We completely agree with this comment of Reviewer 2. In effect, we interpret the data as indicating that some of the effects of EPA on FGF21 are mediated by the GPR120 receptors, but that EPA also has additional mechanisms of action. Indeed, the results we obtained using cells with PPAR α invalidation or PPARY inhibition (see Supplemental Fig 8) indicate that EPA remains active, albeit far less so, for inducing FGF21 in these experimental settings, but the extent of induction of FGF21 by EPA is significantly more moderate. Thus, part of the effects of EPA can indeed be mediated by pathways other than the GPR120, potentially including PPAR-dependent pathways. We have modified the Results and Discussion section to better clarify this point in the revised manuscript. Regarding the differences among the n-3 PUFAs, EPA has been reported to be especially powerful for inducing the browning of WAT in vivo, and our data (and others, e.g., Zhao et al. Biochem. Biophys Res Comm 2014, 450, 1446) have confirmed this response in cell culture. Although EPA, ALA and DHA all reportedly bind GPR120 with high affinity, DHA appears to have a somewhat lower binding affinity relative to the two other n-3-PUFAs (Muyamoto et al. Int J Mol Sci 2006, 17,450). This may explain why DHA did not have clear-cut effects in our brown/beige adipocyte system. Other yet-unknown aspects, such as the specific transport and metabolic processing of DHA versus EPA and ALA in brown and beige adipocytes might also have influenced our findings.

- 4) Page 16-17 at the end of result description, authors have mentioned that GPR120 activates FGF21 and then leads to browning. However, it's not only FGF21, as browning/beiging through activation of GPR120 in FGF21 KO mice is still present, just a bit reduced (Figure 10). So it clearly implies that there are other pathways. What can be the other pathways that GPR120-mediated browning/beiging beside of FGF21. Since FGF21 is not the only one factor for GPR120-mediated browning/beiging effects, authors should suggest or discuss this point.

We thank the Reviewer for this relevant comment, and agree with the points raised. We made sure that our revised manuscript does not over-state the involvement of FGF21. We agree that while the induction of FGF21 is involved in the effects of GPR120 activation on browning, FGF21 cannot account for all the observed effects. In the

revised text (and in also response to the following point of this Reviewer) we now report that treatment of brown adipocytes with GW9508 or EPA up-regulates the activity of various regulatory kinases including p38 MAPK, which is a known inducer of relevant thermogenic (UCP1) and browning-orchestrating (PGC-1a) genes (Collins et al., Mol Endocrinol 2004, 18, 2123). P38 MAPK affects FGF21 expression, but this pathway could also easily account for some direct (i.e. FGF21-independent) effects of GPR120 activation on brown adipocyte activation and browning. This possibility is now discussed in the revised manuscript.

- 5) Overall, how does GPR120 activation induce FGF21 level to regulate browning and beige phenotype? The authors fail to show the mechanism how this might happen. All the data in this study is quite descriptive and repetitive with different depots of adipose tissue without direct molecular mechanism.

Our study was not intended to achieve a precise description of the intracellular action mechanisms of GPR120 activation in brown adipocytes (in fact, these aspects are not totally clear even in other cell systems in which the effects of GPR120 activation have been extensively studied). However, we agree that improving this aspect of our study could enhance the overall strength of our observations.

Accordingly, we now provide new data in the revised manuscript (Fig 2) showing, using specific inhibitors, that the effect of GPR120 on FGF21 gene expression requires the activities of ERK and PI3 kinase (as seen in other cell systems). We also observed the involvement of p38 MAPK. P38 MAPK is considered a major intracellular mediator of thermogenic activation in brown adipocytes as well as a regulator of FGF21 expression (Cao et al., Mol Cell Biol 2004, 24, 3057; Hondares et al, J Biol Chem 2011, 286, 12983). Recently, it has been proposed that GPR120 activation induce p38 MAPK in other systems (Gao et al., 2015; Sci Rep. 5:14080) Here we found that GW9508 or EPA induced p38 MAPK phosphorylation in brown adipocytes. We also observed that GPR120 activation induces transcription from the FGF21 gene promoter, and our deletion analysis revealed that a previously known p38 MAPK-responsive site at the proximal promoter region of the FGF21 gene is required for the effects of GPR120 activation on FGF21 gene transcription. Finally, we found that co-transfection of cells with a dominant-negative form of MKK6 (MKK6-K82A), which is immediately upstream of p38 MAPK, impaired the GPR120 activation-dependent induction of the FGF21 gene promoter. These new data are presented in Results, Figure 3 and Supplemental Fig 9. Overall, our findings suggest that p38 MAPK mediates at least in part the effects of GPR120 activation on FGF21 gene expression. Future work will be needed to precisely define the entire network of intracellular kinases involved in all phases of the response, from GPR120 activation to the targeting of gene transcription.

- 6) Although authors claimed that the negligible expression of GPR40 in BAT would justify their using of GW9508 in the set of studies, using GW9508 in FGF21 KO mice (Figure 9) is not appropriate model in the context of authors' study which they want to emphasize the link between GPR120 and FGF21. GW9508 in animal model will hit GPR40 with stronger activity (for EC50, ~47nM for GPR40 vs ~2.2 uM for GPR120) than to activate GPR120. Although authors described the absence of FGF21 impaired the capacity of GW9508 (in other words GPR120 activation, in the authors' point of view), this data is not clear enough to draw line

between GPR120 and FGF21. Authors should use GPR120 specific agonist (TUG891 or compound A from the recent GPR120 agonist publications), not GW9508 for in vivo study, unless authors generate GPR120 and FGF21 double KO mice to prove the link between GPR120 and FGF21.

We agree that the usefulness of GW9508 is limited in vivo. To further support our conclusion that GPR120 is involved in the response to GW9508 (and also in response to Reviewer 1 and Reviewer 3) we have added a new set of experiments in which GPR120-KO mice were treated with GW9508. We found that key events in the GW9508 response were impaired in GPR120-KO mice: there was much less induction of oxygen consumption, the inductions of FGF21 expression and plasma levels were almost completely abrogated, and the browning of iWAT was reduced. We believe that these data strongly support the involvement of GPR120 in the actions of GW9508 in vivo.

Concerning the use of other drugs in vivo, we initially set out to perform complementary pilot study using TUG891. We found, however, that TUG891 intake caused a marked anorexigenic response in mice (this was also observed in the only previously published study we found to have used oral TUG891; Gozal et al. Int J Obes (Lond). 2016 ,40:1143). Given the strong effects of fasting itself on brown/beige activity and FGF21 levels, this would impair our ability to precisely interpret any obtained results. Thus, we did not continue this experimental model. It is possible that the other drug suggested by the Reviewer, compound A, would be a better alternative for further research. However, this new GPR120 agonist has only very recently (within the past few months) been commercially available. If using compound A in vivo is an editorial requirement for publication, we will perform the necessary experiments. However, we respectfully suggest that the above described experiments using GPR120-KO mice are sufficient to indicate that GPR120 contributes to the GW9508 response in vivo.

7) Although the (online) methods are very detailed and informative, the legends of the main figures could use additional description so that the reader does not have to refer back and forth between sections. Authors also need to carefully label the subset of figures with their description in each figures legend. For example, figure 8 legend is very difficult to figure out what to what. If the legend can be modified like this, it will be much understandable without again "refer back and forth between sections".

Example for modification of figure 8 legend; For a-d, iBAT precursors from wild-type and GPR120-null mice were differentiated (n=3-5). (a) Representative optical microscopic images (magnification 10X). Relative mRNA expression levels of FGF21, UCP1, FABP4 and leptin (b). Effects of EPA on FGF21 mRNA expression and FGF21 secretion (c and d). For e and f, iWAT precursors from wild-type and GPR120-null mice were differentiated into beige adipocytes (n=3-5). Representative optical microscopic images (magnification 10X) (e). Effects of EPA on FGF21 mRNA expression and FGF21 protein secretion (f). (g) Differentiated brown and beige adipocytes were treated with GW9508 or EPA in the presence or absence of AH7614 (a GPR120 antagonist) for 24 h (n=3). FGF21 mRNA expression and FGF21 protein levels in culture medium. (h) Differentiated brown adipocytes were subjected to siRNA-mediated knockdown of GPR120 (see Methods section) and treated with GW9508 or EPA. mRNA expression levels of GPR120, FGF21 and UCP1 (n=3).

We thank the Reviewer for this suggestion, which we have taken to heart. We have re-written the figure legends accordingly, while also staying within the length limits requested by Nature Communications.

Minor points:

- 1) In Figure 3, 3a shows Leptin, 3b shows AdipoQ in adipogenic genes and Bmp8b is pretty much the highest increased with treatment in 3a, but not in 3b? Looks a bit like pick and choose.

We apologize for our mistake; the presentation of AdipoQ was a clerical error. Our lab systematically measures multiple transcripts, including AdipoQ, when characterizing adipose cells. However, this study involved the systematic analysis of leptin and FABP4 as marker genes of the general adipogenic phenotype. This mistake has been now corrected.

We included Bmp8b because, as we had previously reported, it is a very good marker gene of thermogenic activation in BAT (Whittle et al., Cell 2012, 149,871). However, Bmp8b is practically undetectable in WAT, unless the tissue is undergoing a very strong induction of browning, such as that elicited by chronic cold exposure. This is why we included Bmp8b in the data for BAT but not iWAT (with the exception of the cold exposure experiment shown in Fig 4 of the revised manuscript).

- 2) In Figure 7c and d, where is ALA? What about FGF21m RNA expression? In 7g, where is dilapidated condition in FGF21 mRNA graph and ALA in FGF21 protein graph? Again, looks a bit like pick and choose, not consistent, especially when authors show the "same" graphs within one figure.

We are sorry for this inconsistency, which was introduced when we attempted to simplify the presentation of data. We have now homogenized the presentation of data in Fig 7. In the revised manuscript ALA is shown in Fig 7c and 7d. The FGF21 mRNA was indeed included in Fig 7 panel b in the original manuscript, but we understand that it could have been missed given the heterogeneous presentation of upper versus lower panels of original Fig 7. We amended the relevant panels. The results obtained under the delipidated condition and effects of ALA are now shown now in the new Fig 7g.

- 3) Authors need to check their citations thoroughly. The reference number they cited and actual content they would describe are not matched. For example, line 7 from bottom of page 10, the reference they cited is 31, should be 13 for GPR120 study.

We are grateful to the Reviewer for noticing these errors. We apologize for these mistakes and have thoroughly checked the manuscript to avoid citation errors in the revised version.

Reviewer #3 (Remarks to the Author):

The current manuscript provides the molecular mechanisms by which GPR120 controls brown fat thermogenesis through activating FGF21 expression. The authors provided ample data in mice and cultured cells that reinforce the importance of GPR120 in whole body energy homeostasis by promoting brown and beige fat development. This would be an important addition for the future therapeutic intervention using GPR agonists. Overall the data support the authors' conclusions --- the critical experiment missing in the current manuscript is to show the effect of GW9508 in GPR120 KO mice. This would be crucial to establish the causal link between GW9508 and brown and beige fat in vivo.

We thank the Reviewer for the positive comments and agree with her/his point regarding the relevance of testing the effect of GW9508 in GPR120 KO mice. Accordingly, we have added a new data set obtained in GPR120-KO mice treated with GW9508, as presented in Figure 3 and Supplementary materials of the revised manuscript, and described in our reply below.

1. As described above, it would be critical to examine the extent to which the GW9508 effects on BAT thermogenesis and browning are affected in GPR120 KO mice in vivo.

Figure 3 of the revised manuscript shows our main findings in GPR120-KO mice treated with GW9508.

We found that GW9508 treatment increases oxygen consumption in wild-type mice. On the other hand, GPR120-KO mice had a somewhat lower basal oxygen consumption and GW9508 treatment increased oxygen consumption to a significantly lower extent than that seen in wild-type mice.

The ability of GW9508 treatment to induce FGF21 gene expression in the BAT and iWAT of wild-type mice was totally blocked in GPR120-KO mice, and the induction of iWAT browning was significantly reduced.

These data support the hypothesis that GPR120 is largely required for the effects triggered by the GW9508 treatment.

2. Related to this point, the authors should examine if GW9508 increases whole body energy expenditure in wt and GPR120 KO mice.

In response to this relevant comment, and a similar request made by Reviewer 1, we measured oxygen consumption under basal conditions and in response to the β_3 -agonist CL316,243. The obtained results, which are shown in Figure 3 of the revised manuscript, indicates that GW9508 increases oxygen consumption and enhances the response to CL316,243 in wild-type mice, which is consistent with the signs of BAT recruitment and iWAT browning elicited by GW9508. In contrast, GPR120-null mice showed a mild reduction in oxygen consumption, and an impaired induction of oxygen consumption after treatment with GW9508 both under basal and CL316,243-stimulated conditions.

3. The effects of GW in FGF21 KO mice are intriguing. Now the authors have an opportunity to examine how much of the metabolic improving effects of GW9508, such as improved insulin

resistance, are blunted in FGF21 KO mice under diet induced obesity. This would significantly increase the impact of this paper.

We completely agree with the Reviewer's comment, but it is a rather complex matter to assess the effects of GW9508 in a diet-induced obesity model subjected to FGF21 invalidation. First, a proper design model (timing and dose of GW9508) should be established in high fat-diet fed mice to ensure to reproduce an improvement in systemic metabolism and possibly induction of browning in diet-induced obesity conditions, to be applied later high-fat diet-treated FGF21-KO mice. Moreover, diet-induced obesity itself up-regulates the levels, but not the biological activity, of FGF21. In fact, it is FGF21 action that is mostly altered in diet-induced obesity; this has been proposed to be a "FGF21 resistance" condition, especially in adipose tissues (Fisher et al. Diabetes. 2010, 59:2781; Diaz-Delfin et al. Endocrinology 2012, 153:4238). The proposed experiments could therefore not focus solely on the capacity of GW9508 to act in the presence/absence of FGF21. Indeed, we would need to primarily consider the impact of GW9508 on FGF21 resistance. We would thus need to explore the whole FGF21-responsive system, including the expression/activity of the FGFR1 receptor and especially of β -Klotho. Our current findings suggest that this could be a very promising area of research, especially in light of our previous report that inflammation (a putative down-regulated target of GPR120 activation) is a main factor in FGF21 resistance (Diaz-Delfin et al. Endocrinology 2012, 153:4238). We believe, however, that this study would require a comprehensive assessment of the effects of GPR120 on obesity and inflammation in relation with the FGF21 resistance mechanisms and that this is thus beyond the scope of the current brown/beige adipose-oriented study.

Minor points:

1. The authors should provide GPR120 protein expression in key parts of Fig.2.

We now provide data on GPR120 protein expression in Figure 1 of the revised manuscript (Fig 2 of the previous version) and present results validating the specificity of our immunoblot assay (Supplementary Methods).

2. The authors wish to provide UCP1 protein data in some of the critical figures like Fig.9 and 10.

In accordance with this request which was also made by Reviewer 1, we now include data on the protein levels of UCP1 in Figures 2, 3 and 9 of the revised manuscript).

3. While liver is the major source of serum FGF21, pancreas can contribute to serum FGF21 level. It would be helpful to test if GW9508 regulates FGF21 expression in the pancreas.

As requested, we determined the effects of GW9508 on FGF21 expression in the pancreas. However, we did not observe any significant effect. The data are shown in Figure 2c of the revised manuscript.

REVIEWERS' COMMENTS:

Reviewer #1 (Remarks to the Author):

The authors have thoughtfully and adequately addressed the reviewers concerns. I find this to be an interesting and thorough study.

Reviewer #2 (Remarks to the Author):

This version of the manuscript is substantially improved and many criticisms of the original are addressed.

Although I still think they should do the experiments with GPR120 specific agonist not with dual agonist, GW9508 for GPR40 and GPR120. The diminished effect of GW9508 in GPR120 KO mice is still vague to nail out specific role of GPR120 in BAT related to FGF21, while other tissues, such as liver (main source of FGF21), express GPR40. So it might contribute effect on BAT via systemic regulation.

Reviewer #3 (Remarks to the Author):

The authors addressed the reviewer's comments.

REVIEWERS' COMMENTS:

Reviewer #1 (Remarks to the Author):

The authors have thoughtfully and adequately addressed the reviewers concerns. I find this to be an interesting and thorough study.

We thank the positive appreciation of our revised manuscript by the Reviewer.

Reviewer #2 (Remarks to the Author):

This version of the manuscript is substantially improved and many criticisms of the original are addressed.

Although I still think they should do the experiments with GPR120 specific agonist not with dual agonist, GW9508 for GPR40 and GPR120. The diminished effect of GW9508 in GPR120 KO mice is still vague to nail out specific role of GPR120 in BAT related to FGF21, while other tissues, such as liver (main source of FGF21), express GPR40. So it might contribute effect on BAT via systemic regulation.

We thank the positive appreciation of our revised manuscript by the Reviewer. We agree on the limitation stated by the reviewer and, accordingly, we have stated in the revised manuscript the caution required in the interpretation of experiments using GW9508 "in vivo" in light of the dual agonist properties of the drug (page 8, last paragraph).

Reviewer #3 (Remarks to the Author):

The authors addressed the reviewer's comments.

We thank the positive appreciation of our revised manuscript by the Reviewer